# Optoelectronic properties and ultrafast carrier dynamics of copper iodide thin films

Zhan Hua Li[1,2], Jia Xing He[3], Xiao Hu Lv[1,2], Ling Fei Chi[1], Kingsley O. Egbo[4,5], Ming-De Li [3] ✉, Tooru Tanaka [6], Qi Xin Guo [6], Kin Man Yu[4] & Chao Ping Liu [1,2] ✉

As a promising high mobility p-type wide bandgap semiconductor, copper iodide has received increasing attention in recent years. However, the defect physics/evolution are still controversial, and particularly the ultrafast carrier and exciton dynamics in copper iodide has rarely been investigated. Here, we study these fundamental properties for copper iodide thin films by a synergistic approach employing a combination of analytical techniques. Steady-state photoluminescence spectra reveal that the emission at ~420 nm arises from the recombination of electrons with neutral copper vacancies. The photogenerated carrier density dependent ultrafast physical processes are elucidated with using the femtosecond transient absorption spectroscopy. Both the effects of hot-phonon bottleneck and the Auger heating significantly slow down the cooling rate of hot-carriers in the case of high excitation density. The effect of defects on the carrier recombination and the two-photon induced ultrafast carrier dynamics are also investigated. These findings are crucial to the optoelectronic applications of copper iodide.

P-type transparent semiconductors with controlled hole concentration ($N$) and high hole mobility ($\mu$) are crucial for current optoelectronics. However, obtaining transparent p-type semiconductors with desired optoelectrical properties is extremely challenging. For instance, efficient p-type doping for the common transparent conducting oxides (TCOs) is rather difficult, because their O $2p$ derived valence band maximum (VBM) is too low in energy and the VBM states are highly localized[1–3] resulting in high ionization energies for most acceptors and large hole effective mass or low hole mobility $\mu$. Hence, seeking p-type transparent materials with a high VBM position with respect to the vacuum level as well as a large valence band (VB) dispersion is greatly desired.

With a wide direct bandgap of 3.1 eV, p-type transparent copper iodide (CuI) has attracted increasing attentions in recent years because of its relatively high hole mobility and large exciton binding energy

(~62 meV)[4,5] CuI has a zincblende ground-state phase below 643 K[6] and demonstrated versatility in a wide range of optoelectronic applications, e.g., as efficient hole-transport layer in solar cells[7–10] thin film transistors[11–13] p-n heterojunctions[14–16] thermoelectrics[17,18] and lasing devices[19]. The VBM of CuI mainly consists of Cu $3d$ and I $5p$ orbitals, while its conduction band minimum (CBM) is mostly composed of Cu $4s$ states. Nominally undoped CuI exhibits a hole concentration $N$ in the range of $10^{16}$–$10^{20}$ cm$^{-3}$, mainly attributed to the energetically favored formation of copper vacancy ($V_{Cu}$) acceptors with rather low ionization energy $\varepsilon(0/-)$ of ~44 meV, while compensating donor native defects (e.g., $V_I$, $Cu_I$, and $Cu_i$) have relatively high formation energies as the Fermi level $E_F$ approaches the VBM[20]. In comparison with most TCOs, CuI has a much larger VB dispersion, with a small hole effective mass $m_h^*$~0.3 m$_0$. A record hole mobility $\mu$ of ~43.9 cm$^2$ V$^{-1}$ s$^{-1}$ in single-crystalline CuI has been reported[21], which is two orders of magnitude

[1]Department of Physics, College of Science, Shantou University, 515063 Shantou, Guangdong, China. [2]Center of Semiconductor Materials and Devices, Shantou University, 515063 Shantou, Guangdong, China. [3]Department of Chemistry, Shantou University, 515063 Shantou, Guangdong, China. [4]Department of Physics, City University of Hong Kong, 83 Tat Chee Ave., Kowloon, Hong Kong. [5]Paul-Drude-Institut fur Festkorperelektronik, Leibniz-Institut im Forschungsverbund Berlin e. V, Hausvogteiplatz 5-7, 10117 Berlin, Germany. [6]Synchrotron Light Application Center, Saga University, Saga 840-8502, Japan. ✉e-mail: mdli@stu.edu.cn; cpliu@stu.edu.cn

higher than that of commonly used p-type NiO ($\mu < 0.1\,cm^2\,V^{-1}\,s^{-1}$)[22–24] Appropriate extrinsic acceptor doping (e.g., by substituting the iodine with chalcogen impurities like S or Se) can be employed to further increase the p-type conductivity of CuI[25,26] In the low excitation density (i.e., the density of photogenerated electron-hole pairs less than the corresponding Mott density) regime, the optical properties of CuI are dominated by excitonic transitions. For instance, the absorption spectrum exhibits two sharp peaks at ~3.06 eV ($Z_{1,2}$ excitonic absorption related to the doubly degenerated VBM at the Γ point) and ~3.7 eV ($Z_3$ transition due to the split-off band), respectively[27,28] while two prominent peaks at ~3.06 eV (corresponding to the free-exciton recombination) and ~2.95 eV (related to the bound exciton recombination or donor-acceptor pair recombination) respectively can be observed in photoluminescence (PL) measurements.

Ultrafast carrier dynamics are known to determine the electronic transport and optical properties of semiconductors. In other words, the performance of optoelectronic devices depends strongly on the generation, relaxation and trapping/recombination processes of charged carriers involved. Therefore, understanding these ultrafast transient processes (with time scales from ~100 fs to ~1 ns) is vital to optimize the design and the performance of various optoelectronic devices[29–31] Despite numerous efforts devoted to the characterization and device applications of CuI, the ultrafast carrier dynamics in CuI, which is also closely related to the defects, has rarely been investigated[32].

In this paper, we report on a comprehensive study of optoelectronic properties and ultrafast carrier dynamics of CuI thin films by using a variety of analytical techniques including Hall-effect measurements, spectroscopic ellipsometry (SE), PL spectroscopy and femtosecond transient absorption (fs-TA) spectroscopy. The p-type transparent CuI thin films were synthesized by iodization of $Cu_3N$ thin films. Effects of excitation density and defects on the carrier dynamics were investigated in terms of the corresponding time scales and the mechanisms for theses ultrafast processes (e.g., thermalization, cooling, recombination, etc.).

## Results
### Crystal structure and surface morphology
Figure 1a shows the grazing incidence x-ray diffraction (GIXRD) spectra of CuI thin films with different treatments (i.e., as-grown, PTA in air, PTA in $I_2$). The diffraction peaks observed in Fig. 1a can be indexed to pure CuI (JCPDS, 06-0246). The CuI thin films exhibit diffractions from the (111), (220), and (311) planes, consistent with previous reports[16]. The grain sizes were calculated by Scherrer equation from the widths of the (111) diffraction peak. For CuI with thickness ~100 nm, the grain size ($D$) for film with post-growth thermal annealing (PTA) in air ($D$~37.1 nm) and $I_2$ ($D$~32.5 nm) is larger than that of the as-grown ($D$~30.5 nm) sample. Apparently, such annealing treatments in air or $I_2$ can enhance the crystallinity of CuI thin film. In order to investigate the dependence of full width at half maximum (FWHM) of XRD peak or the

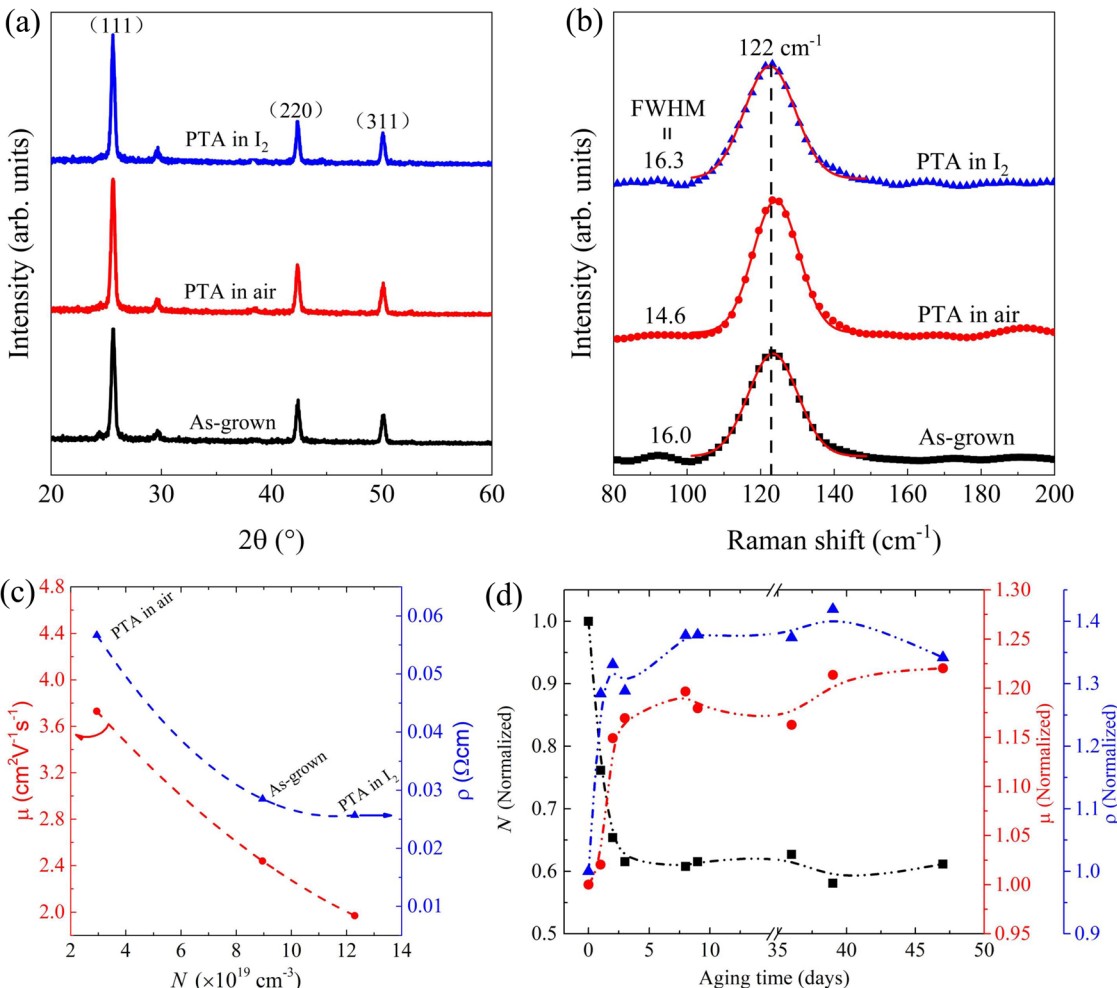

**Fig. 1 | Structural and electrical properties.** GIXRD spectra (**a**) and Raman spectra (**b**) of CuI thin films with different treatments, namely, as-grown (black), PTA in air (red), and PTA in iodine vapors (blue). The red lines in **b** are the corresponding peak fits. **c** Electrical properties of CuI thin films with different treatments (i.e., as-grown, PTA in air, PTA in $I_2$). **d** Normalized electrical properties of as-grown CuI thin films (on glass substrate) with increasing aging time. In **c** or **d**, the black square, red circle and blue triangle denote the $N$, $\mu$ and $\rho$, respectively, and the corresponding dash lines as guides for eye.

grain size on the film thickness, we further carried out XRD measurements for the as-grown CuI thin films with different thicknesses. The FWHM of the (111) diffraction peak is found to decrease from 0.308° to 0.119°, corresponding to grain size increasing from 26.1 to 67.7 nm, as the film thickness increases from ~60 to ~300 nm (see Supplementary Fig. 1). Therefore, the crystallinity of as-grown CuI thin film improves with film thickness. In the following sections, results were taken from film with thickness of ~100 nm, unless otherwise specified. The lattice parameter of the as-grown CuI thin film is found to be 6.05 Å, which is close to the value from the literature[16]. Note that there are no noticeable diffraction peaks from $Cu_3N$ phase, suggesting that the films were completely iodized. Raman spectroscopy was further used to study the vibrational modes of the CuI thin films. Figure 1b shows the Raman spectra of CuI thin films with different treatments. The main feature is the peak at 122 cm$^{-1}$ which can be identified as the transverse optical (TO) mode[33], while the relatively weak longitudinal optical (LO) peak (~140 cm$^{-1}$) is probably masked by the broad TO peak[34]. Although the Raman spectra of the three films look similar, a non-negligible broadening of the TO mode peak can be obvious for films with different treatments. In Fig. 1b, the FWHM for the TO peak ranging from 14.6–16.3 cm$^{-1}$ is indicated for each film. Such broadening of the TO peak can be attributed to the presence of disorders and/or defects in CuI[33]. The TO peak of CuI thin film becomes narrower after PTA in air, which is most likely due to the increased grain size and thus reduced disorder. In contrast, the relatively wide TO peak observed for the CuI films as-grown and PTA in $I_2$ atmosphere may be due to their relatively high concentration of acceptor defects (e.g., iodine interstitials $I_i$, $V_{Cu}$ etc). The excess I may be incorporated as interstitials or promote Cu vacancy formation. The SEM image shown in Supplementary Fig. 2a reveals that the as-grown CuI thin film includes sharply defined triangular grains, with relatively rough surface and high density of voids. The corresponding AFM image is depicted in Supplementary Fig. 2b, with a rms roughness of ~12 nm. The effects of PTA treatments on the surface morphology of CuI thin films are negligible. Be noted that compact CuI thin film with smoother surface (rms roughness < 5 nm) have been reported using other growth methods (e.g., pulsed laser deposition[35]).

## Electrical properties

Figure 1c shows the hole concentration ($N$) dependent hole mobility ($\mu$) and the resistivity ($\rho$) of CuI thin films with different treatments. The as-grown CuI thin film exhibits $N$~$9 \times 10^{19}$ cm$^{-3}$, $\mu$~2.4 cm$^2$ V$^{-1}$ s$^{-1}$, and $\rho$~$10^{-2}$ $\Omega$ cm, close to values reported previously[36]. It is observed that PTA in air or $I_2$ has significant effect on the electrical properties of the CuI thin film. Specifically, the $N$ of CuI thin film after PTA in air ($I_2$) decreases (increases) to ~$3 \times 10^{19}$ cm$^{-3}$ (~$1.2 \times 10^{20}$ cm$^{-3}$), with the corresponding mobility $\mu$ increases (decreases) to ~3.8 cm$^2$ V$^{-1}$ s$^{-1}$ (~2 cm$^2$ V$^{-1}$ s$^{-1}$). Among all the native defects in CuI thin film either grown in Cu-rich or I-rich conditions, the copper vacancy $V_{Cu}$ has the lowest formation energy and ionization energy, and thus mainly responsible for its p-type conductivity[16]. Nevertheless, the electrical properties of CuI thin film should also be related with other defects involved. It was reported that the formation energies of these native acceptors (i.e., $V_{Cu}$, $I_i$ and $I_{Cu}$) in CuI in an I-rich environment become lower[20]. This implies that the CuI thin film after PTA in $I_2$ may have a higher concentration of native acceptor defects, consistent to our experimental observations and other reported results[37]. It is worth mentioning that partial contribution of holes from the $I_i$ cannot be ruled out, since the hole concentration of CuI thin film after PTA in $I_2$ underwent a fast reduction with aging time and the $I_i$ might be shallow acceptors as indicated from first-principles calculations[20]. However, the nature (i.e., shallow or deep acceptor) of $I_i$ is still controversial. For instance, some studies found that the $I_i$ are deep acceptors with ionization energy ~0.5 eV[16,38] in contrast to the shallow acceptors as revealed in other investigations[20]. On the other hand, the reduction in $N$ for the CuI thin film upon PTA in

air is most likely due to the possible removal and/or annihilation of $V_{Cu}$ and out-diffusion of $I_i$. In addition, the formation of additional compensating donor defects (e.g., $V_I$, $Cu_i$, defect cluster $V_I + Cu_i$) in the Cu-rich condition[38] cannot be ruled out. In order to evaluate the stability of the as-grown CuI thin film on glass substrate, the normalized electrical properties were recorded upon aging in the ambient air with increasing aging time, as shown in Fig. 1d. It is found that the hole concentration decreases rapidly to a stabilized value of ~60% of its original $N$ with a corresponding increase in the mobility by ~20% within three days, which is also likely due to the reduced concentration of $V_{Cu}$ and $I_i$ upon aging. This suggests that the high $N$ in the as-grown samples comes from a high concentration of acceptor defects and ~40% of these defects are very mobile with the rest (60%) are more stable at RT. The more mobile defects are likely $I_i$ while the more stable ones are $V_{Cu}$. Regarding the high $N$ in the nominally undoped CuI, possible contributions from the oxidation and incorporation of atmospheric oxygen into the CuI thin film were proposed recently by Storm et al.[35]. However, in our experiments, the $N$ of CuI thin films did not show any noticeable enhancement upon aging in ambient air. Based on DFT calculation, Grauzinyte et al.[25] found that due to its high formation energy, the substitutional oxygen in the iodine sublattice (i.e., $O_I$) have a rather low equilibrium defect concentration (~$10^6$ cm$^{-3}$) at room temperature. Therefore, we believe that the hole concentration in our CuI thin films is mainly attributed to their native defects (e.g., $V_{Cu}$ and or $I_i$) rather than the possible incorporation of O.

## Optical properties

The optical properties of CuI thin films were investigated by using SE. SE analysis was based on the three-layer optical model, i.e., the glass substrates, CuI thin film, and a surface roughness layer. For CuI thin films, a Psemi-M0 oscillator, a Tauc-Lorentz oscillator, three Gaussian oscillators and a Drude oscillator were used to describe their dielectric functions. These dielectric function models can fit the experimental SE data ($\Psi$, $\Delta$) quite well over the whole spectral range. Figure 2a, b shows the measured and the fitted $\Psi$ and $\Delta$, respectively for the as-grown CuI thin film with the angle of incidence of 65°. Be noted that index grading was used for the CuI layer in the optical model to describe the linear variation of refractive index throughout the film thickness induced by its graded microstructure as frequently observed in transparent conducting thin films (e.g., tin doped indium oxide)[39]. It is found that the variation of optical properties from the bottom of the film to the film surface is quite low (e.g., the change in the refractive index $\Delta n/n$ < 5%, while the change in the absorption coefficient $\Delta\alpha/\alpha$ < 6.4%) in the spectral range of 3–4 eV. Such small change in the index would not affect the main conclusion. In the following context, the mean value of the optical constants (e.g., dielectric function, refractive index n, absorption coefficient $\alpha$) of the thin film were used for calculations and discussions. We also find that the derived dielectric function of as-grown CuI thin film (see Supplementary Fig. 3a) is quite close to that of CuI film with similar hole density (~$1.6 \times 10^{19}$ cm$^{-3}$) as determined numerically by a Kramers−Kronig consistent point-by-point regression analysis[28]. The absorption coefficient ($\alpha$, obtained from SE analysis) of the CuI thin film with different treatments are shown in Fig. 2c. The excitonic band edge absorption at $E_0$~3.06 eV (denoted as $Z_{1,2}$) and the absorption peak at $(E_0 + \Delta_0)$~3.7 eV (denoted as $Z_3$) are distinct for the CuI thin film with PTA in air, while they both almost disappear for the as-grown and the CuI thin film with PTA in $I_2$. This suggests that the excitonic effects in the CuI thin film are negligible as the carrier density is larger than ~$8 \times 10^{19}$ cm$^{-3}$, due to the screening of Coulomb interaction. The free carrier absorption (at photon energies < 1.2 eV) is notable for the CuI thin films with relatively high hole density (> $8 \times 10^{19}$ cm$^{-3}$), as seen from Fig. 2c. In comparison with the as-grown CuI thin film, the CuI thin film with PTA in $I_2$ shows negligible change in $\alpha$, while the CuI thin

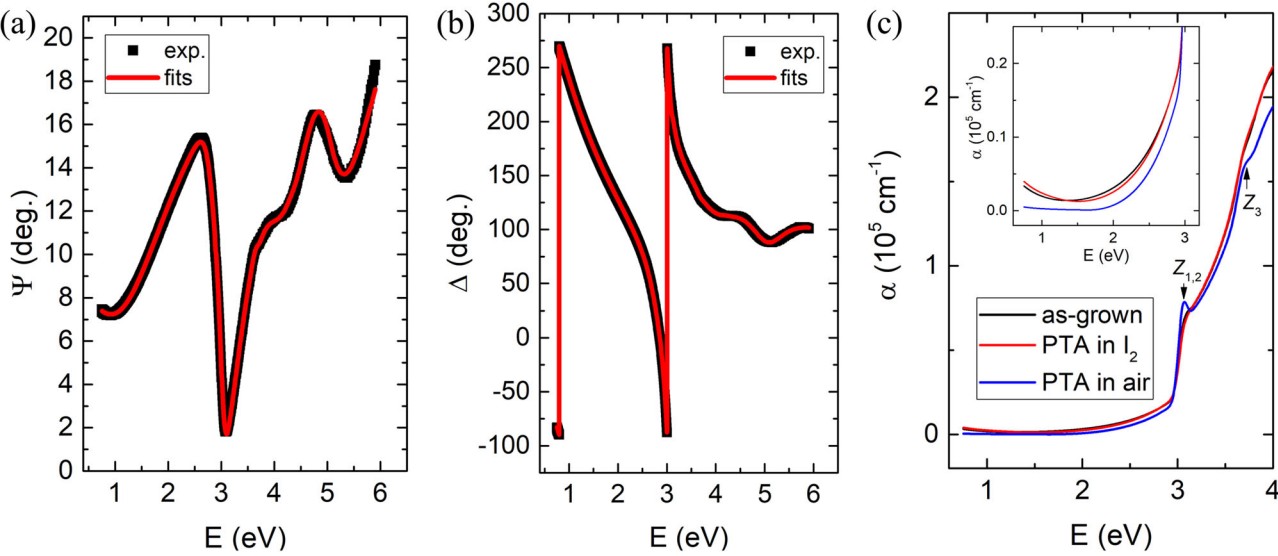

**Fig. 2 | SE spectra and the derived absorption coefficient.** The measured (black square) and the fitted (red line) SE spectra ($\Psi$ and $\Delta$ in **a** and **b**, respectively) of the as-grown CuI thin film with the angle of incidence of 65°. **c** Absorption coefficient $\alpha$ of CuI thin film with different treatments. Inset: zoom of $\alpha$ in the low value region.

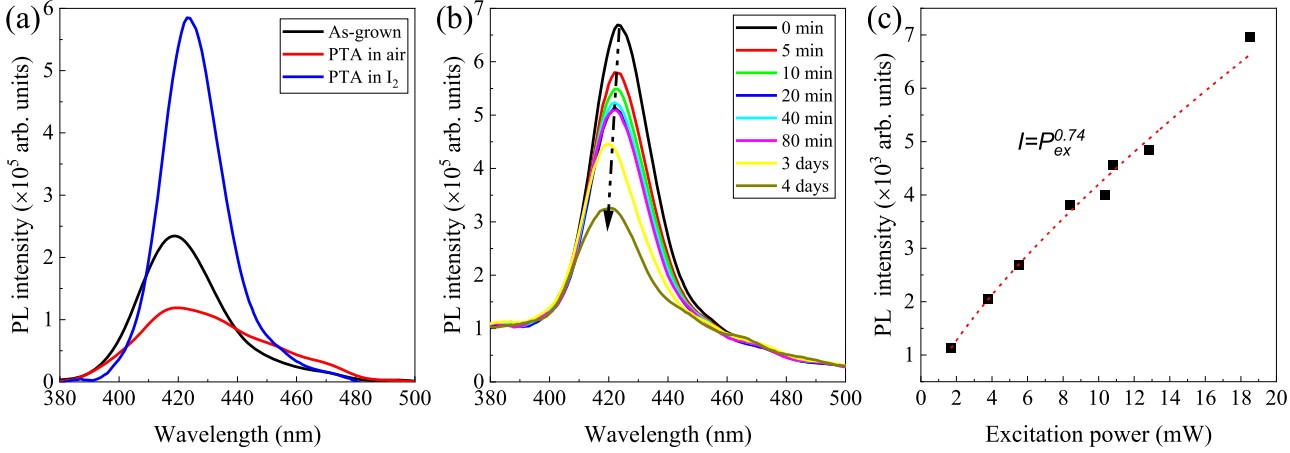

**Fig. 3 | Steady-state PL emissions. a** Steady-state PL spectra of CuI thin films with different treatments, i.e., as-grown (black), PTA in air (red) and PTA in $I_2$ vapor (blue). **b** PL intensity of as-grown CuI thin film with different aging time in ambient air. **c** Excitation power ($P_{ex}$)-dependent PL intensity of as-grown CuI thin film, with the fits denoted as the red dashed line.

film with PTA in air shows noticeable reduction in sub-gap absorption as well as the free carrier absorption (<1 eV). The origin of the sub-gap absorption is related with native defects (e.g., $V_{Cu}$, $V_I$, and $V_I + Cu_i$ defect clusters etc) and/or the defects at grain boundaries. We also note that excessive $I_2$ adsorption on the film surface and at the grain boundaries may be present in the case of relatively high concentration of $I_2$ vapor in the PTA treatment[40], which was verified by its slightly yellowish color as well as its high sub-gap absorption coefficient. Nevertheless, the exact origin of color in CuI is still quite controversial. For example, Gao et al. ascribed this coloration of CuI to adsorbed iodine species on the crystal surface[41], while Koyasu et al. suggested that possible native defects (e.g., $V_{Cu}$, $V_I$, and $I_{Cu}$) in the bulk crystal rather than the surface states may account for its unique optical properties[42]. Since native defects in CuI are rather unstable, the yellowish color from our fresh CuI thin film with PTA in $I_2$ may also be related to the evolution of native defects involved. The average visible transmission ranges from 60 to 80%, which is higher than that of most p-type TCOs (e.g., NiO) with a similar film thickness[1] but much higher resistivity ($\rho$~0.1–1 $\Omega$ cm). In addition,

the CuI thin film deposited on a flexible PET substrate using the same process exhibits similar optoelectronic properties as that of on glass and sapphire substrates, suggesting its potential applications in flexible optoelectronic devices.

PL measurements were also employed on the CuI films to further reveal details of the defects. Figure 3a shows the steady-state PL spectra of these CuI thin films. A broad PL peak centered at ~420 nm (i.e., 2.95 eV) is observed for these CuI thin films, which is most likely due to the recombination of the electrons in the conduction band with the neutral $V_{Cu}$ (i.e., $V_{Cu}^0$)[42–45] denoted as (e, $V_{Cu}^0$) recombination. In this scenario, the higher concentration of $V_{Cu}$ would provide more states (unfilled) for (e, $V_{Cu}^0$) recombination and thus higher radiative recombination probability at ~420 nm. In comparison with the as-grown sample, the CuI thin film with PTA in air/$I_2$ vapor exhibits a reduced/enhanced PL intensity, which can be mainly ascribed to the enhanced/reduced trap-assisted nonradiative recombination. It is worth mentioning that the PL quantum efficiencies of these CuI thin films are rather low, indicating the dominance of nonradiative recombination (e.g., Shockley-Read-Hall (SRH) recombination and/or

Auger recombination[46]). Upon aging in the ambient air, the PL intensity of as-grown CuI thin film is found to decrease as the aging time increases (see Fig. 3b), which is probably resulted from their gradual increase/decrease in the concentration of deep defects (e.g., $V_I$)/shallow acceptors (e.g., $V_{Cu}$). Interestingly, the PL peak presents a slight blue-shift from 423 to ~420 nm (with energy difference ~21 meV) as increase of aging time. In order to better understand this PL peak shift, the optical properties of CuI thin films with $N$ ranged from $2.79 \times 10^{17}$ to $1.68 \times 10^{20}$ cm$^{-3}$ (achieved by PTA in air at 100 °C for varying durations) were studied by using SE. The optical bandgap $E_G^{opt}$ was estimated from the zero crossing of the first-derivative of the $\varepsilon_2$ spectra, $d\varepsilon_2/dE$[28]. As the hole density increases from $2.79 \times 10^{17}$ to $1.68 \times 10^{20}$ cm$^{-3}$, the $E_G^{opt}$ of CuI thin film increases from ~3.057 to ~3.11 eV, as shown in the Supplementary Fig. 3d. As $N < ~5 \times 10^{19}$ cm$^{-3}$, the obtained $E_G^{opt}$ approximately equal to the corresponding free-exciton peak energy (see the green region of Supplementary Fig. 3d), while as $N > ~7 \times 10^{19}$ cm$^{-3}$, the $E_G^{opt}$ is close to its band edge absorption energy (see the red region of Supplementary Fig. 3d). This implies that exciton peak energy blue shifts as the hole density in CuI thin film increases, mainly due to the enhanced Coulomb screening induced reduction in exciton binding energy ($E_b$), similar to that observed in anatase TiO$_2$[47]. The energy difference between the valence band edge $E_V$ and the Fermi level $E_F$ (i.e., $E_V$-$E_F$) was calculated by using $N = 2\frac{m_h k_B T}{2\pi \hbar^2}^{3/2} F_{1/2}(-\frac{E_F - E_V}{k_B T})$, where $N$ is the hole density in the valence band, $m_h$ is the density of state hole effective mass ($m_h = (m_{hh}^{3/2} + m_{lh}^{3/2})^{2/3}$) which is found to be around 2.47 $m_e$ for CuI[16], $k_B$ is the Boltzmann constant, T is the temperature, $\hbar$ is the Plank constant, $F_{1/2}$ is the Fermi-Dirac integral which was calculated by method proposed by Bednarczyk et al.[48]. It is found that $E_V$-$E_F$ is around 0, and 29 meV for samples with hole density $N$-7.6 $\times 10^{19}$ and $1.68 \times 10^{20}$ cm$^{-3}$, respectively. Given the similar band dispersions (e.g., $m_e^* = m_h^* = 0.3m_0$) for the CBM and the VBM of CuI[16], the Burstein-Moss (BM) shift ($\Delta E_{BM}$) due to the hole filling in the valence band of CuI thin film is approximately equal to the $2(E_V - E_F)$, i.e., $\Delta E_{BM} ~2 (E_V - E_F)$, indicating that the $\Delta E_{BM}$ of CuI thin film with $N$-1.68 $\times 10^{20}$ cm$^{-3}$ is ~58 meV. For CuI thin film with negligible exciton resonance, the optical bandgap can be expressed as

$$E_G^{opt} = E_G + \triangle E_{BM} - \triangle E_{BGR} \quad (1)$$

where $E_G$ is the intrinsic bandgap of 3.1 eV, $\Delta E_{BGR}$ is the bandgap narrowing due to the bandgap renormalization (BGR). It is found that the $\Delta E_{BGR}$ for CuI thin film with $N$-1.68 $\times 10^{20}$ cm$^{-3}$ is ~48 meV, which is larger than the energy shift (i.e., 21 meV) as observed in the PL spectra (Fig. 3b). This is understandable, since the 48 meV represents the upper limit of $\Delta E_{BGR}$ for our CuI thin films, while the energy shift of ~21 meV is related to the change in the $\Delta E_{BGR}$ (i.e., $\Delta(\Delta E_{BGR})$) for the sample with increasing aging time. Since the $\Delta E_{BGR}$ is determined indirectly from Eq. (1), quantification of the BGR effect in CuI thin films with distinct excitonic resonance (i.e., samples with $N < ~7.6 \times 10^{19}$ cm$^{-3}$) is not achieved in the present work. The right $Y$-axis of Supplementary Fig. 3d further illustrates the quantity of $E_b'$ (i.e., $E_b' = 3.1 - E_G^{opt}$) for rough estimates of exciton binding energy $E_b$ as a function of $N$ with the assumption of $\Delta E_{BGR} = 0$. Apparently, $E_b'$ should be slightly larger than the $E_b$ ($E_b$3.1 $- \triangle E_{BGR} - E_G^{opt}$), and the negative value of $E_b'$ indicates the negligible excitonic resonance. It is worth mentioning that optical absorption edge is related with the effects of BGR and BM shift, while PL peak is not affected by the BM shift. The independence of PL peak on the BM shift of CuI thin film was revealed in our experiments. As seen in Fig. 3b, the PL spectra are dominated by the (e, $V_{Cu}^0$) recombination in the CuI thin films with hole densities up to ~10$^{20}$ cm$^{-3}$, without any noticeable effect from the BM shift in the degenerate samples. This can be caused by following two reasons, (i) fast nonradiative recombination (e.g., Auger recombination, SRH recombination) of the majority photogenerated carriers as discussed in the later sections; (ii) efficient radiative (e, $V_{Cu}^0$) recombination process related to the

photogenerated electrons relaxed to the CBM by fast scattering and the prominence of $V_{Cu}^0$ defect states. These two types of ultrafast recombination (i.e., nonradiative and radiative) processes are probably unrelated with the BM shift in the CuI thin films. Nevertheless, upon variation in concentration of defect (e.g., $V_{Cu}$) densities, the band tails of CuI thin film and/or the broadening of the corresponding defect band may also change and thus cause the PL peak shift. Hence, the PL peak shift with aging time as observed in Fig. 3b may be attributed to the change in the BGR effect and/or the variations in the band tails/broadening of the defect band in the CuI thin film[49]. We also note that there is much debate on the origin of emission at ~420 nm. For instance, Nikitenko et al. assigned the PL peak at ~420 nm to donor-acceptor pair (DAP) recombination[16]; Nakamura et al. found that the CuI thin film grown by molecular beam epitaxy exhibits a prominent PL peak due to the free-exciton (denoted as $X$) recombination at ~3.06 eV (405 nm) at RT, while this $X$ emission peak gradually disappears and another PL peak at ~2.96 eV (~420 nm), attributed to the bound exciton recombination, becomes distinct with decreasing temperature ($T < 150$ K)[27]. It is unlikely that the PL peak at ~420 nm for our CuI thin films at RT arises from the bound exciton recombination, since the bound exciton typically has a rather small exciton to the impurity or defect binding energy[16]. It is worth mentioning that the origin of the PL emission at ~420 nm from the band-to-band recombination cannot be ruled out due to the possible band-tail states induced PL emission (i.e., Stokes shift). To further elucidate the nature of the PL emission at ~420 nm, Fig. 3c illustrates the PL intensity as the function of excitation power of the as-grown film. The power-dependent PL intensity can be roughly expressed by using the expression $I$-$P_{ex}^k$, where $I$ is PL intensity, $P_{ex}$ is the excitation power and $k$ is an exponent in range of 0–2[50]. Generally, $k = 2$ indicates the recombination of free electrons with holes, $1 < k < 2$ corresponds to free or bound exciton decay, while $k < 1$ is related with impurity or defect-related emission. By fitting the PL intensity, the $k$ is found to be ~0.74, further verifying that the observed emission at ~420 nm can be ascribed to the defect assisted emission, i.e., (e, $V_{Cu}^0$) recombination. Given the distinct excitonic absorption as well as the broad PL emission peak as observed in the as-grown CuI thin film (see Figs. 2, 3), the presence of free excitonic emission (at ~406 nm) cannot be ruled out.

## Photocarrier generation and recombination kinetics

Pump-probe fs-TA measurements were carried out to understand the charge carrier relaxation, trapping and recombination processes in these CuI thin films. The 2D color plot of TA spectra of the as-grown CuI thin film with $\lambda_p = 320$ nm and $F_p = 0.64$ mJ/cm$^2$ is shown in Fig. 4a. For comparison, the absorption (black line) spectrum is also depicted. The TA spectrum presents two negative (i.e., $\Delta A < 0$) photobleaching (PB) peaks at ~406 nm (3.05 eV) and ~336 nm (3.7 eV), corresponding to the photon energies for the free-exciton transitions (i.e., $Z_{1/2}$ and $Z_3$ in Fig. 2). In general, free e-h pairs rather than excitons would be directly generated in semiconductors upon photon excitation with energy greater than the bandgap ($E_G$). Subsequently, the photoexcited carriers relax to the band edges by phonon emission (via carrier-phonon scattering) and lead to the formation of bound e-h pairs (excitons). Before their recombination or under a steady-state excitation, a thermodynamic equilibrium between the excitons ($X$) and the e-h plasmas (EHP) can be established, i.e., $X \rightleftharpoons e + h$. It is worth mentioning that the formed excitons (if any) can be transformed into an e-h plasma by strong collisions (between the excitons) induced-dissociation and/or by the thermal ionization resulting from the strong screening of excitons. At equilibrium, the fraction of free carriers relative to the excitation density $x$ (i.e., $x = N_e/(N_e + N_X) = N_e/N_0$) is related with the Saha equation[51]

$$\frac{x^2}{1-x} = \frac{1}{N_0}(2\pi m_r k_B T/h^2)^{3/2} \exp\left(-\frac{E_b}{k_B T}\right) \quad (2)$$

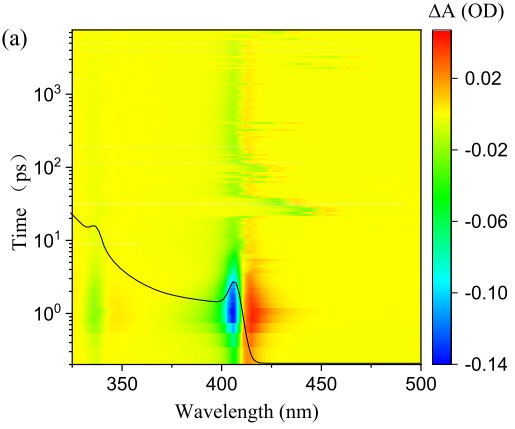
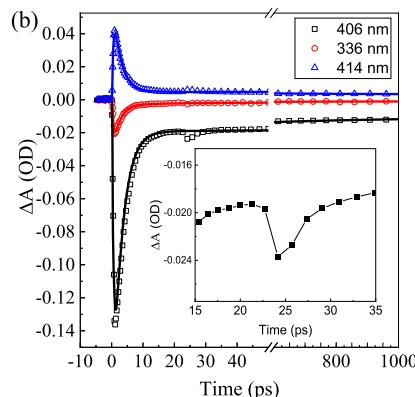

**Fig. 4 | TA spectra and kinetic traces. a** 2D color plot of TA spectra of as-grown CuI thin film under the pump excitation with $\lambda_p = 320$ nm and $F_p = 0.64$ mJ/cm². The corresponding absorption (black line) spectrum is also shown for comparison. **b** Kinetic traces in the first 1 ns at selected wavelengths. Inset: kinetic at 406 nm zoomed-in with time delay in the range of 15–35 ps. All the spectra are for the as-grown CuI thin film.

where $N_0$ is the excitation density, $m_r$ is the exciton reduced effective mass, $k_B$ is the Boltzmann constant, $h$ is the Planck constant, $T$ is temperature, and $E_b$ is the exciton binding energy. Be noted that the $E_b$ depends on the excitation density $N_0$ due to the Coulomb screening and phase-space filling effects, which can be expressed by ref. 52

$$E_b(N_0) = E_{b0}\exp(-N_0/N_{\text{Mott}}) \tag{3}$$

where $E_{b0}$ is the exciton binding energy at low carrier concentration and $N_{\text{Mott}}$ is the critical Mott density. For CuI, $E_{b0}$ is ~62 meV[16], while the Mott density can be calculated by $N_{\text{Mott}} = k_B T / 11\pi a_\chi^3 E_{b0}$, yielding $N_{Mott}$ of ~$4.5 \times 10^{18}$ cm⁻³ at 300 K (using the exciton Bohr radius $a_\chi = 1.39$ nm). Based on Eq. (3), the $E_b(N_{Mott})$ is found to be ~24 meV, implying that excitons cannot be stable at RT due to the thermal ionization as the photogenerated carrier density is $> N_{Mott}$. Note that the Saha relation in Eq. (2) is only valid at relatively low excitation densities ($N_0 < N_{Mott}$) with negligible many-body effects[51]. The concentration of charge-carriers excited by the pump excitation is governed by the absorption coefficient (at $\lambda_p$) of the material as well as the $F_p$. The concentration of excited carriers (i.e., the excitation density $N_0$) can be approximated by using the equation[53,54]

$$N_0 = \frac{F_p \cdot \alpha_z}{E_p}\left(\frac{2}{1+n_z}\right)^2 \tag{4}$$

where the squared factor takes into account the reflective losses at the air-sample interface, $F_p$ is the pump fluence, $E_p$ is the pump photon energy, $\alpha_z$ and $n_z$ are the absorption coefficient and refractive index at the $\lambda_p$, respectively. For $F_p$~0.13 mJ/cm² and 1.27 mJ/cm² ($\lambda_p$~320 nm), the $N_0$ is found to be ~$1.0 \times 10^{19}$ cm⁻³ and $1.0 \times 10^{20}$ cm⁻³, respectively, which are both higher than the corresponding $N_{Mott}$(~$4.5 \times 10^{18}$ cm⁻³) by 1–2 orders of magnitude. Generally, the excitons would cease to exist as individual quasi-particles and thus excitonic effect is negligible as the photogenerated carrier density $> N_{Mott}$. However, the exciton absorption is distinct in CuI thin film with $N$~$5.0 \times 10^{19}$ cm⁻³ (around one order of magnitude higher than the calculated Mott density), as indicated in Fig. 2c and Supplementary Fig. 3. The persistence of such excitonic feature at carrier density above the Mott density is most likely due to the formation of Mahan excitons, which were also reported in other materials with large exciton binding energies[47]. As shown in Supplementary Fig. 3b, as the hole density is $> $~$7 \times 10^{19}$ cm⁻³, the exciton absorption becomes negligible. Be noted that a fraction of EHP may be present in the photogenerated carriers. By using Eq. (2), the fraction $x$ is found to be ~88% and 57% for photogenerated carrier density ~$1 \times 10^{17}$ and $1 \times 10^{18}$ cm⁻³, respectively. Therefore, the PB at 406 nm (or 336 nm)

(see Fig. 4a, with excitation density ~$5.0 \times 10^{19}$ cm⁻³) for the CuI thin film can be attributed to the formation of EHP, intraband carrier relaxation, as well as the filling of excitonic states. In addition, a positive $\Delta A$ at longer wavelengths is also observed, which can be ascribed to the photo-induced absorption (PIA) resulting from the BGR effect[55,56] In order to better understand the kinetics of PB and BGR, Supplementary Fig. 4a, b illustrate the TA spectra at several selected time delays ranged from 0.2 ps to 2 ps with $F_p = 0.64$ mJ/cm² and $F_p = 2.54$ mJ/cm², respectively. As seen, both the PB at 406 nm (or 336 nm) and the BGR occur at a very short time scale of <200 fs after excitation, indicating the ultrafast processes of EHP formation and the subsequent thermalization in the CuI film. Under pump fluence of 0.64 (2.54) mJ/cm², the change in $\Delta E_{\text{BGR}}$ due to the transient BGR effect reaches its maximum value of ~10 (18) meV at a time delay of ~1 (1.2) ps, which was estimated from the energy shift at the transition point (i.e., PIA-PB transition point, see Supplementary Fig. 4). In addition, the major PB peak exhibits a blue-shift of ~8 meV with increasing $|\Delta A|$ under a certain pump fluence (e.g., 0.64 mJ/cm²). In case high carrier density filled in the band edges, long-range Coulomb screening may result in a blue-shift of the exciton peak (if excitons are present) due to the reduced exciton binding energy, while phase-space filling may also contribute to the blue-shift of exciton peak or the absorption edge[47], which is actually revealed in Supplementary Fig. 4. A high energy tail (with $\Delta A < 0$) is also observed in the TA spectrum above the $E_g$ in the wavelength range of 350–390 nm. Such high energy tails in the TA spectra might result from the quasi-equilibrium carrier distribution (i.e., the thermalized distribution) at a relatively high temperature[55,57–60] Before the thermalization, the distribution of photocarriers is nonthermal (i.e., the nonthermal regime, in which the distribution function cannot be described by the temperature dependent Fermi-Dirac function), occurring at a time delay typically less than 200 fs.

To better understand the carrier dynamics, the $\Delta A$ signals recorded at the two maxima of the bleach (i.e., 406 and 336 nm) and the photo-induced absorption at 414 nm are shown in Fig. 4b and fitted using following equation[61],

$$S(t) = e^{-\left(\frac{t-t_z}{t_p}\right)^2} * \sum_i A_i e^{\frac{-(t-t_z)}{t_i}} \tag{5}$$

$$t_p = \frac{IRF}{2ln2} \tag{6}$$

Here, IRF is the instrument response function, $t_p$ is the FWHM of the IRF, $t_z$ is the time zero for chirp correction, while $A_i$ and $t_i$ are amplitudes and the time constant or the lifetime, respectively,

* stands for convolution. As shown in Fig. 4b, the transient ΔA signals for the three selected wavelengths can be fitted to a model with a monoexponential rise with the time constant $\tau_1$ and a biexponential decay with time constants $\tau_2$ and $\tau_3$, respectively. The fast rise (with $\tau_1$-500 fs) of the ΔA at the PB maxima can be mainly attributed to the thermalization (via carrier-carrier scattering) and the initial cooling (via carrier-phonon scattering) of photogenerated carriers in the CuI thin film. While the fast decay process with a time constant of $\tau_2$-3 ps is primarily related to the carrier trapping and nonradiative recombination. Since the photogenerated carrier density is relatively high ($>10^{19}$ cm$^{-3}$) in our experiments, the nonradiative Auger recombination would be significant. Hendry et al. found that the Auger recombination takes place directly at a time delay of ~1.5 ps from the hot EHP (before their cooling) with a density above the corresponding $N_{Mott}$ in ZnO[53]. Therefore, the fast decay process with a time constant $\tau_2$-3 ps is very likely due to the prominent Auger recombination of the EHP, while the relatively slow decay process with a time constant of $\tau_3$-650 ps may be related to the electron-hole recombination, denoted as (e, h) recombination. The recombination mechanisms are further verified by an analysis using the rate equation in the following paragraph. It is also interesting to find that the ΔA undergoes a slight rise at a time delay of ~25 ps in the decay process with a decay time at around 3.68 ps, as illustrated in the Fig. 4b and its inset. We note that such phenomenon only appears in the case of relatively high pump fluence ($F_p > $~0.1 mJ/cm$^2$). Wille et al. reported lasing emission from a CuI microwire under fs-pulsed excitation with a threshold density of 0.3 mJ/cm$^2$[19]. Given the high excitation density ($>N_{Mott}$) under a pump fluence of 0.3 mJ/cm$^2$, the stimulated emission as observed in the CuI microwire was most likely due to the radiative recombination in a degenerate EHP. In addition, the characteristic decay time for the laser emission from the CuI microwire was found to be in range of 3.5–8.5 ps, which is close to the decay time (~3.68 ps) observed from the inset of Fig. 4b. Therefore, it is reasonable to conjecture that the slight rise in ΔA at $\tau_D$-25 ps owes to the stimulated emission (from the radiative recombination in a degenerate EHP) induced re-excitation in the CuI thin film.

To better elucidate the recombination kinetics in the CuI thin film, fs-TA measurements with different $F_p$ and $\lambda_p$ were also carried out. Figure 5a displays transient absorption kinetic traces of the PB peak at ~406 nm for the as-grown CuI thin film under various $F_p$ from 0.13 to 2.54 mJ/cm$^2$ ($\lambda_p = 320$ nm). For a rough estimation, the photogenerated/excess carrier density $n$ at a specific time delay would be proportional to the transient absorption change, i.e., $n \propto |\Delta A|$, and thus the $n$ in the decay process can be calculated using the excitation density. Qualitatively, the kinetics of TA can be explained by the interplay between different carrier recombination mechanisms, namely the trap-assisted SRH recombination, exciton recombination, band-to-band (bimolecular) recombination and third-order Auger recombination, which can be approximately captured by the following rate equation[62],

$$-\frac{dn}{dt} = An + Bn^2 + Cn^3 \tag{7}$$

where the density of free carriers $n$ is a function of the time delay $t$, and $A$, $B$, and $C$ are the recombination coefficients related with the SRH recombination and/or exciton recombination (first-order, mono-molecular recombination), bimolecular recombination (second-order), and Auger recombination (third-order), respectively. Assuming one single recombination mechanism, the solution for the first-, second-, and third-order kinetics are expressed as

$$\ln n - \ln n_0 = -A(t - t_0) \tag{8}$$

$$\frac{1}{n} - \frac{1}{n_0} = B(t - t_0) \tag{9}$$

$$\frac{1}{n^2} - \frac{1}{n_0^2} = 2C(t - t_0) \tag{10}$$

where $n_0$ and $n$ are the excess carrier density at time delay of $t_0$ and $t$, respectively. If the recombination of carriers is dominated by a single mechanism (i.e., SRH, exciton, bimolecular or Auger) at a time delay ranging from $t_0$ to $t$, the major recombination mechanism can be distinguished from the relationship between $n$ and $t$, as described in Eq. (8–10). However, this assumption may overestimate the corresponding recombination coefficient, since other recombination processes (e.g., trap-assisted SRH process) may also contribute slightly or even partially to the overall recombination. Thus the obtained recombination coefficients by using Eq. (8–10) correspond to their upper limits. Be noted that the Auger recombination would be dominant only in the case of high carrier density (e.g., $N_0 > 10^{19}$–$10^{20}$ cm$^{-3}$). Accordingly, the Auger recombination coefficient $C$ is only extracted from cases with high excitation density (e.g., $N_0 > 10^{19}$ cm$^{-3}$), while the SRH recombination coefficient $A$ is derived from cases with relatively low excitation density (e.g., $N_0 < 10^{19}$ cm$^{-3}$). Figure 5b, c plots the $\ln(n)$-$\ln(n_0)$ and the $1/n^2 - 1/n_0^2$, respectively at 406 nm as a function of $(t-t_0)$ with different pump fluences. The $t_0 = 1.4$ ps in Fig. 5b, c is the time delay corresponding to $|\Delta A|$ rising to its maximum. It is found that $\ln(n)$-$\ln(n_0)$ shows a linear dependence of time delay as the $F_p$ is relatively low (i.e., < 0.64 mJ/cm$^2$), indicating that the initial decay process (with t < 10 ps) is dominated by the trap-assisted SRH recombination. In contrast, the $1/n^2 - 1/n_0^2$ presents a linear relationship with time delay less than 6 ps as the $F_p$ is relatively high (e.g., >1 mJ/cm$^2$), suggesting that the corresponding decay process is governed by the Auger process. Interestingly, we also find that the decay process can last for a long time >7000 ps, as shown in Fig. 5d. In addition, the $1/n - 1/n_0$ presents a linear dependence as time delay is >30 ps, which can be attributed to the radiative (e, h) recombination process. It is worth noting that the aforementioned possible stimulated emission from the degenerate EHP likely occurs in the initial stage of this (e, h) recombination process. From their linear fits as shown in Fig. 5b–d, the corresponding (average) recombination coefficients (i.e., $A$, $B$, $C$ for the SRH, bimolecular and Auger process, respectively) can be derived, with $A$-$2.42 \times 10^{11}$ s$^{-1}$, $B$-$1.30 \times 10^{-11}$ cm$^3$ s$^{-1}$, and $C$-$7.42 \times 10^{-30}$ cm$^6$ s$^{-1}$. The inset of Fig. 5d is the plot of $1/n - 1/n_0$ vs. $(t - t_0)$ under relatively low pump fluence (i.e., 0.13 mJ/cm$^2$), which gives $B$ (~$10^{-9}$ cm$^3$s$^{-1}$) by ~2 orders magnitude larger than that of from cases with high pump fluences (e.g., 2.54 mJ/cm$^2$). By using Eq. (2, 3), the fraction of exciton is found to be ~20% for the excess carriers with density ~$2 \times 10^{17}$ cm$^{-3}$, which implies that the high bimolecular recombination coefficient $B$ extracted from the cases with low pump fluences might be overestimated owing to partial contribution from the exciton recombination. It is worth mentioning that the recombination coefficients in semiconductors are highly dependent on their excess carrier densities[52,63] which is also manifested in Fig. 5b–d, i.e., different slopes or recombination coefficients under different excitation densities. For the cases with high excitation density (e.g., >$10^{19}$ cm$^{-3}$), the excitonic process would be negligible at the initial decay process with carrier density >$10^{20}$ cm$^{-3}$.

Effects of pump fluence $F_p$ on the maximum signal ΔA are also investigated. Figure 6a shows TA spectra of CuI thin film with different $F_p$ ($\tau_D = 5$ ps, $\lambda_p = 320$ nm). As shown in Fig. 6b, the amplitude of ΔA at

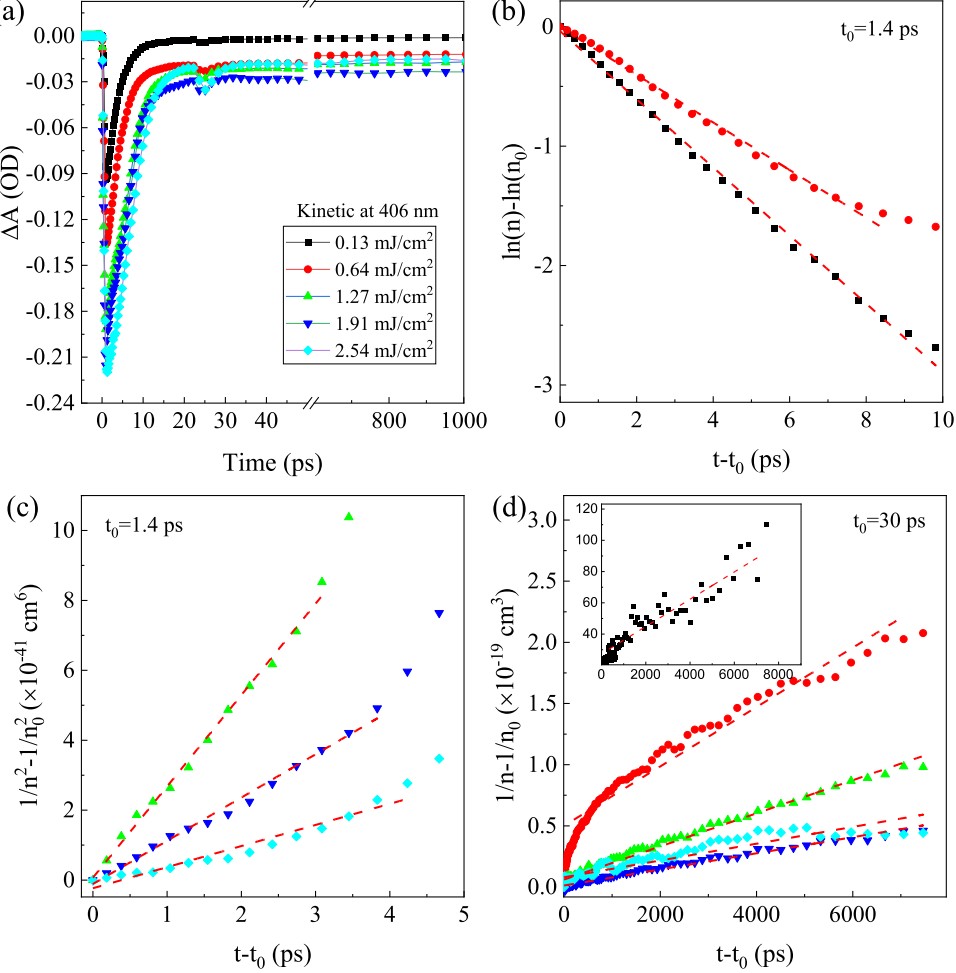

**Fig. 5 | Analysis of recombination kinetics. a** Kinetic at 406 nm with different pump fluence. **b** $\ln(n) - \ln(n_0)$ vs. $(t - t_0)$. **c** $1/n^2 - 1/n_0^2$ vs. $(t - t_0)$, and **d** $1/n - 1/n_0$ vs. $(t - t_0)$ with different pump fluences, with the inset corresponding to this plot under 0.13 mJ/cm². The red dash lines represent their linear fits. In

**a**–**d**, different symbols denote corresponding data obtained under different pump fluences, i.e., 0.13 mJ/cm² (black square), 0.64 mJ/cm² (red circle), 1.27 mJ/cm² (green up triangle), 1.91 mJ/cm² (blue down triangle), and 2.54 mJ/cm² (cyan diamond).

the bleaching maxima of 406 nm increases linearly with the pump fluence with $F_p < 1.5$ mJ/cm², and saturates at a constant value of ~−0.22 when $F_p$ is > ~1.5 mJ/cm². Such saturation of photogenerated carriers was also observed in other semiconductors, e.g., $CH_3NH_3PbI_3$, GaN, and ZnO[64–66] which was attributed to the EHP induced band filling (i.e., Burstein-Moss shift) and even the exhaustion of the number of available states under high excitation fluence. In contrast to the TA spectra under relatively low $F_p$, the TA spectra exhibits a notable absorption at photon energies larger than bleach energies (i.e., 406 and 336 nm) when the $F_p > 2$ mJ/cm² (see Fig. 6a). Such photo-induced non-resonance absorption at energies above or below the bandgap was also observed in monolayer $WS_2$ when excited by an intense laser pulse[67], which was ascribed to the strong photo-induced restructuring of energy bands.

It is worth mentioning that the accurate determination of transient absorption may also need to take into account the transient reflectance as well, though the change in reflectivity can be neglected for low refractive index materials. In order to verify the effect of reflectivity change on the overall absorption change, we performed the transient transmission and transient reflection measurements with incident angle of probe beam of 45° (see the Supplementary Fig. 5 for the schematic diagram). By only taking into account the reflection at the air/CuI interface, the transient absorption coefficient change $\Delta\alpha$

can be expressed as,

$$(-\triangle\alpha)l = \frac{\triangle T}{T} + \frac{R}{1-R}\frac{\triangle R}{R} \tag{11}$$

where $l$ is the path length of the transmitted light in the CuI thin film, $\frac{\triangle T}{T}$ is the relative transmission change, $\frac{\triangle R}{R}$ is the relative reflection change, $R$ is the reflectance (incident angle of 45°) at the steady-state (i.e., without pump excitation) which can be derived from the SE analysis. The derivation of Eq. (11) can be found in the Supplementary Note 1. Supplementary Fig. 6a shows the relative transmission change $\Delta T/T$, relative reflectivity change $\Delta R/R$, and $(-\Delta\alpha)l$ at $\lambda_{probe} = 406$ nm as function of time delay ($\lambda_p = 320$ nm, $F_p = 2.54$ mJ/cm²). Supplementary Figure 6b plots the relative absorption coefficient change $\Delta\alpha/\alpha$ at $\lambda_{probe} = 406$ nm under pump fluences of 2.54 mJ/cm² and 1.91 mJ/cm². As seen, the maximum reduction in $\alpha$ at this wavelength is -55%, i.e., $|\Delta\alpha/\alpha|$-55%, implying that the complete bleaching of exciton absorption occurs as the (photogenerated) carrier density reaches ~$2 \times 10^{20}$ cm⁻³. Supplementary Fig. 7a shows the relative contribution to the absorption change, i.e., $(\frac{R}{1-R}\frac{\triangle R}{R})/(\frac{\triangle T}{T})$, as a function of time delay (<15 ps) at three typical wavelengths (i.e., 395, 406, and 414 nm). As seen, the consequence of neglecting the change in reflection ($\Delta R$) depends on the wavelength, e.g., (i) at the wavelength of 406 nm,

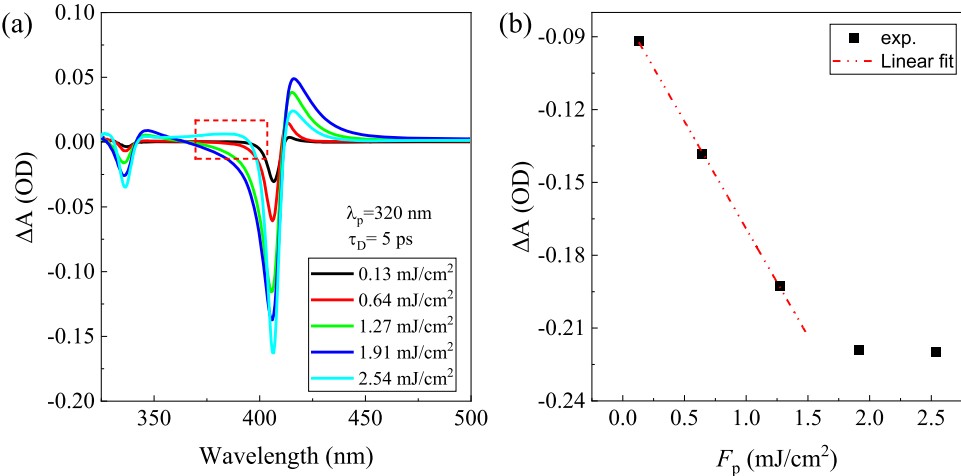

**Fig. 6 | Dependence of TA spectra on pump fluence. a** TA spectra of CuI film under different $F_p$ ($\lambda_p$ = 320 nm) at a time delay of 5 ps. **b** The maximum of ΔA vs $F_p$ monitored at 406 nm.

$|(\frac{R}{1-R}\frac{\triangle R}{R})/(\frac{\triangle T}{T})|$ is less than 0.05, indicating that ΔR of CuI thin film has negligible influence in evaluation of the transient absorption ΔA at this probe wavelength, (ii) at the longer wavelength of 414 nm, $(\frac{R}{1-R}\frac{\triangle R}{R})/(\frac{\triangle T}{T})$ presents relatively high positive values, up to -1.25 at time delay ~3.3 ps, indicating the under estimation of ΔA, iii) at the shorter wavelength of 395 nm, the $(\frac{R}{1-R}\frac{\triangle R}{R})/(\frac{\triangle T}{T})$ is negative, with its maximum absolute value less than -0.3, implying a little over estimation of ΔA. Supplementary Fig. 7b further plots the α of CuI thin film at different time delays, with the $\alpha_0$ representing the absorption coefficient without pump excitation. As seen, the α is reduced at wavelength of 406 nm, while it is increased at wavelength of 395 or 414 nm, which is consistent with the results from Fig. 6a, i.e., PB at ~406 nm and PIA at 395 or 414 nm. A direct comparison of the transient absorption spectra without (ΔT/(T + R)) and with ((ΔR + ΔT)/(T + R)) consideration of transient reflection change (ΔR/(T + R)) at a time delay of 2 ps are depicted in Supplementary Fig. 7c. Again, it is found that the effect of ΔR on the transient absorption spectra is not significant. Therefore, we can conclude that neglecting the reflection change ΔR in our TA measurement would not qualitatively change the results. Thus, in the following sections the TA spectra are derived only from their transient transmission spectra.

## Hot carrier cooling dynamics

Figure 7a further illustrates the normalized TA spectra under different pump fluences ($F_p$ < 2 mJ/cm², $\lambda_p$ = 320 nm) at the time delay of 5 ps. Owing to the increased band filling with increasing $F_p$[55], the PB peak in TA spectra shows a slight blue-shift (with energy difference ~10 meV), meanwhile the high energy tails also become more prominent. In the initial cooling stage, carrier-phonon scattering plays the dominant role. Under high pump fluence, the cooling of hot carriers may slow down due to the effect of hot-phonon bottleneck, i.e., the excess hot-phonon population increases the phonon reabsorption and thus reduce the net cooling rate[68,69]. Since hot carriers occupy electronic states with relatively higher energies above the bandgap energy (i.e., 406 nm), it is preferable to study the decay kinetics of ΔA at a higher energy (e.g., 395 nm) in order to better understand the effect of hot-phonon bottleneck in the CuI thin film. Figure 7b plots the decay kinetics of TA spectra probed at wavelength of 395 nm with different $F_p$ as well as their fits by using Eq. (5, 6). Interestingly, the decay signal recorded at 395 nm can be fitted to a monoexponential decay model, with the time constant (corresponding to the major cooling process) increasing from ~1 to ~3.5 ps as the $F_p$ increases (see Supplementary Table 1). Such increase in the time constant with the $F_p$ is due to the effect of hot-phonon bottleneck. Based on a model calculation,

Herrfurth et al. found that the relaxation of hot carriers in CuI is dominated by carrier-LO-phonon interaction and the hot-phonon effect can delay the relaxation process by few picoseconds[70], which is in good agreement with our findings.

To better elucidate the hot carrier cooling dynamics, the carrier temperature as a function of time delay is calculated. The high energy tail of TA above the band edge (from 3.1 to 3.4 eV) reflects energy distribution of state filling by hot carriers. Once the thermalization is reached for the EHP, the carrier temperature can be estimated by fitting the TA spectra above the band edge in a certain energy range (e.g., 3.1–3.4 eV for our case as shown in Fig. 7c) using the relation $|\triangle A| \propto \exp((E_f - E)/k_B T_C)$, where $E_f$ is the quasi-Fermi level, $k_B$ is the Boltzmann constant, $T_C$ is the carrier temperature. Figure 7d shows the $T_C$ for the as-grown CuI thin film under different pump fluences (with $\lambda_p$ = 320 nm). For relatively low $F_p$ (i.e., ~0.13 mJ/cm²), the initial $T_C$ cools down rapidly from ~700 K to ~RT within 3 ps. As the $F_p$ increases, the initial $T_C$ becomes higher with an increased cooling time constant $\tau_C$ due to the effect of hot-phonon bottleneck. For instance, $\tau_C$ is larger than 4 ps for relatively high $F_p$ (e.g., >0.64 mJ/cm²). Note that the obtained $\tau_C$ is larger than the time constant as given in Supplementary Table 1, which can be ascribed to their different cooling processes included, i.e., the former process is related to carrier-phonon scattering and the phonon-phonon scattering, while the latter is mainly associated with the carrier-phonon scattering. With a high pump fluence, the $T_C$ also exhibits a slight rise at a time delay of ~10 ps with the corresponding $\tau_C$ > 6 ns, as indicated in Fig. 7d. Similar with the ΔA as shown in the inset of Fig. 4b, such slight rise in $T_C$ can be attributed to the aforementioned stimulated emission induced re-excitation, while the relatively long $\tau_C$ most likely results from the Auger heating when the photogenerated carrier density is high[58].

## Two-photon induced ultrafast carrier dynamics

Two phase-coherent photons with $h\nu < E_g$ can be absorbed cooperatively in semiconductors, and thus the electron-hole pairs can be generated by such two-photon absorption (2PA) process. 2PA has been found in different semiconductors (e.g., ZnO, β-Ga₂O₃, GaAs etc)[71,72]. For instance, two-photon and three-photon absorptions in ZnO were attributed to the defect-states-mediated multiphoton absorption process[72]. In order to investigate the possible 2PA in the CuI thin film, the fs-TA measurements were also carried out with the pump excitation energies less than the $E_G$ of CuI. Figure 8a shows the TA spectra of CuI film under different $F_p$ with $\lambda_p$ = 520 nm and $\tau_D$ = 0.5 ps. As seen, the magnitudes of both the high energy tail (due to the band filling) and the low energy tail (due to the PIA) increase as the $F_p$ increases,

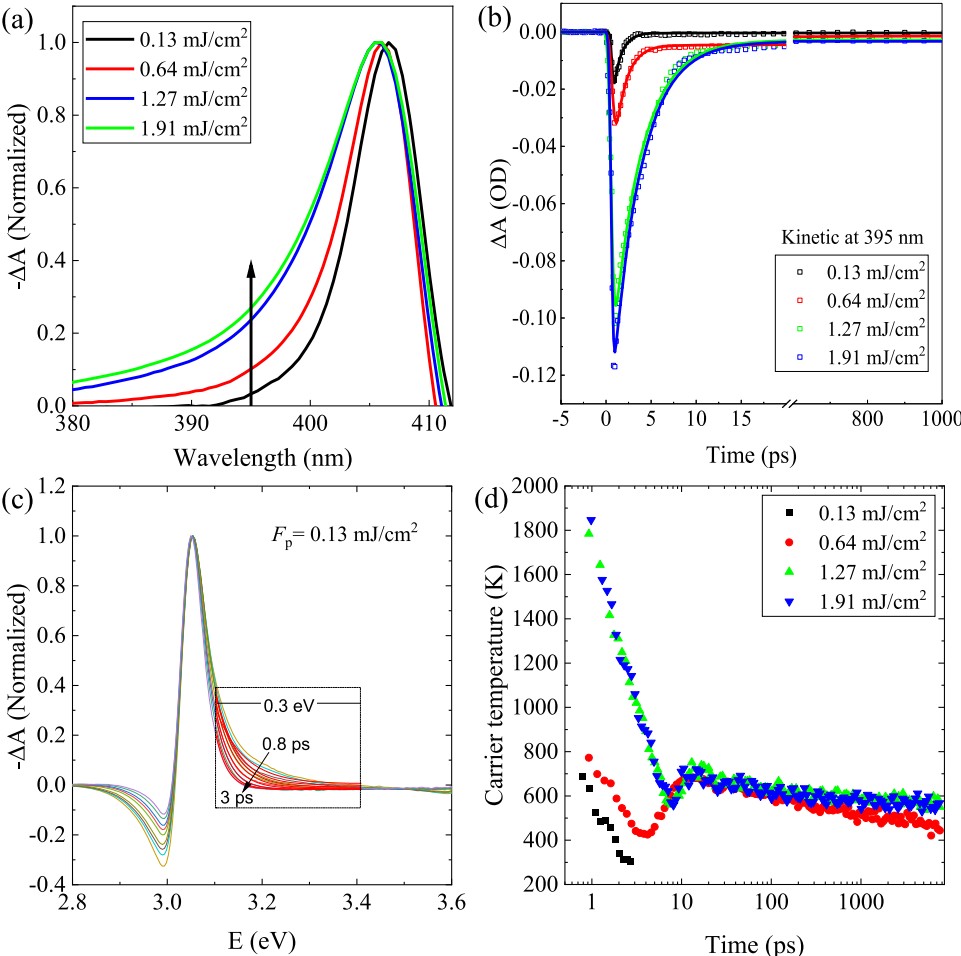

**Fig. 7 | Pump fluence dependent hot carrier cooling dynamics. a** Normalized TA spectra around the peak at 406 nm under different pump fluences with $\lambda_p = 320$ nm. **b** Kinetic profiles of 395 nm bleach under different pump fluences. **c** Normalized TA spectra at different time delays ($F_p$-0.13 mJ/cm$^2$), with the high energy tails (between 3.1 and 3.4 eV) fitted by an exponential function corresponding to Maxwell–Boltzmann distribution to extract the hot carrier temperature $T_C$. **d** The obtained $T_C$ as a function of time delay.

analogous to the TA spectra under excitation energy $> E_G$. Similar phenomena are also observed for TA spectra under pump excitation with longer $\lambda_p$ (e.g., 600 nm). 2PA is a third-order nonlinear process and is highly dependent on the intensity of the excitation beam, in which the attenuation is proportional to the square of the laser intensity[73]. The measured $|\Delta A|$ signal (recorded at 5 ps after excitation) shows a linear dependence with the square of pump fluence $F_p$ (see the inset of Fig. 8b), further verifying the photoexcitation via 2PA process. The closed-aperture Z-scan measurement was also performed to investigate the 2PA process. As illustrated in Fig. 8c, the corresponding Z-scan trace shows a distinct valley at $z = 0$, confirming the existence of 2PA.

Figure 8b shows the kinetic traces probed at 406 nm with different pump fluences, and the decay kinetics are fitted by a mono-exponential decay model, with the results given in Supplementary Table 2. The corresponding time constant or lifetime is found to increase with $F_p$. Be noted that the carrier dynamics under the pump excitation with energy $< E_G$ (as shown in Fig. 8) differs significantly from that of under pump excitation with energy $> E_G$ (as shown in Fig. 5), which is mainly due to their large difference in the excitation densities ($N_0 \propto |\Delta A|_{max}$). When the excitation density is close to the Mott density, the formation of excitons and thus the co-existence of excitons and EHP are very likely to occur in the recovery process. The estimated excitation density under excitation ($\lambda_p = 520$ nm) with $F_p = 0.26 \sim 2.6$ mJ/cm$^2$ is in the range of $4 \times 10^{17} - 8 \times 10^{18}$ cm$^{-3}$, which is

below or close to the Mott density for CuI. Hence, we may infer that the bleaching at 406 nm is due to the filling of exciton levels by the induced exciton population in these cases with relatively low carrier density ($< 10^{18}$ cm$^{-3}$). To further reveal the recombination of the 2PA induced carriers, Fig. 8d plots the $\ln(n) - \ln(n_0)$ as a function of time delay with different pump fluences ($\lambda_p = 520$ nm). Interestingly, two different types of monomolecular recombination can be identified from their initial decay processes (time delay < 20 ps), as indicated by the two different linear regions. The corresponding (average) recombination coefficients extracted from the first linear region (with their fits denoted by the red dash lines) is $A_1$- $7.93 \times 10^{11}$ s$^{-1}$, which is close to the SRH recombination coefficient obtained from the TA spectra with higher excitation densities (see Fig. 5b). Hence, the decay process in the initial 5 ps is dominated by the trap-assisted SRH recombination. While the other monomolecular recombination coefficient derived from the second linear region (with their fits denoted by the black dash lines) is $A_2$-$6.40 \times 10^{10}$ s$^{-1}$, one order of magnitude smaller than $A_1$, which is most likely related to the exciton recombination. This ultrafast carrier trapping (~ ps) occurring in CuI after photoexcitation is comparable with that in ZnO[74].

In Supplementary Fig. 8, the different recombination coefficients (i.e., SRH recombination coefficient $A_1$, exciton recombination coefficient $A_2$, bimolecular recombination coefficient $B$, and Auger recombination coefficient $C$) for the as-grown CuI thin film with varying photogenerated carrier densities are summarized. Apparently, the

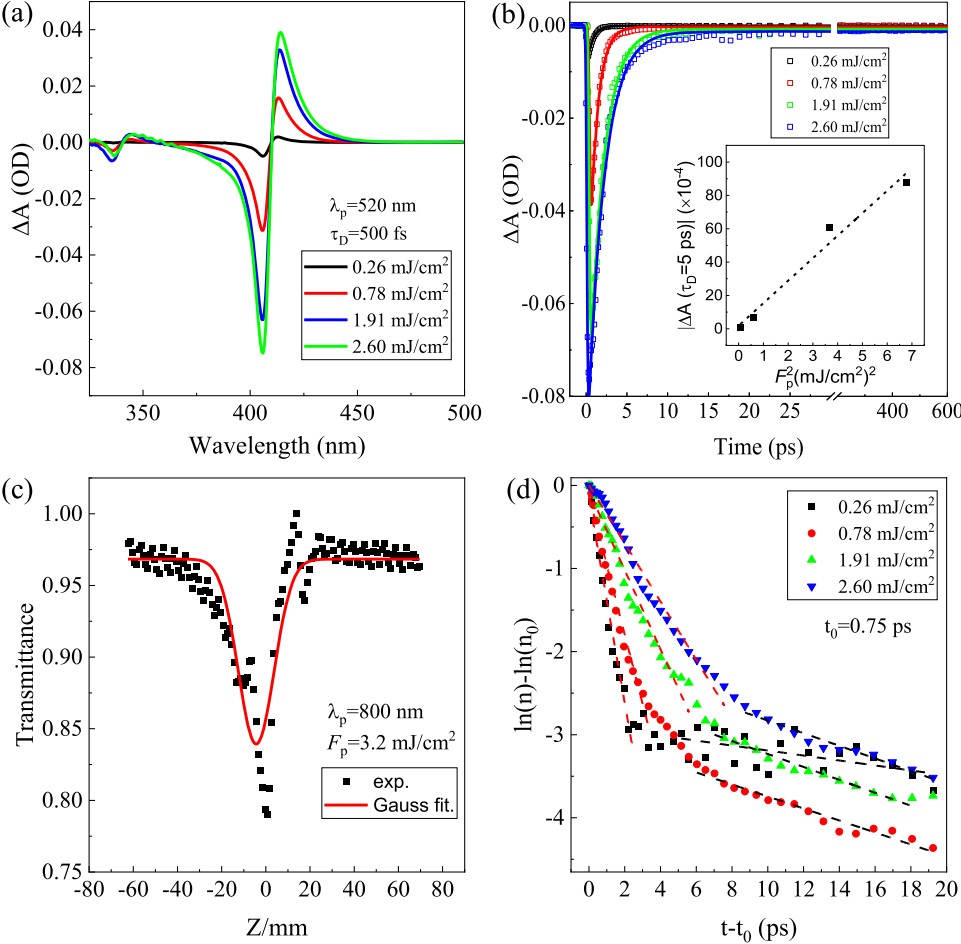

**Fig. 8 | Ultrafast carrier dynamics in two-photon absorption. a** TA spectra of as-grown CuI thin film under different pump fluences at a time delay of 0.5 ps with $\lambda_p = 520$ nm, and **b** corresponding kinetic traces at 406 nm with the best fitting (solid line). Inset in **b** shows the measured $|\Delta A|$ signal (recorded at 5 ps) with quadratic scaling of the pump pulse fluence $F_p$ with the linear fit (dotted line). **c** Z-scan of the as-grown CuI thin film under excitation with $\lambda_p = 800$ nm and $F_p = 3.2$ mJ/cm². **d** $\ln(n) - \ln(n_0)$ vs. $(t - t_0)$ plots under different pump fluences ($\lambda_p = 520$ nm). The dash lines represent their linear fits.

recombination coefficients (i.e., $A_1$, $B$, and $C$) show reduction as increasing of carrier density, consistent with the observation from other semiconductors (e.g., GaN)[52,63] The lifetime of SRH recombination $\tau_{SRH}$ can be written as

$$\tau_{\text{SRH}} = \tau_{\text{min}} + \tau_{\text{maj}} \frac{n}{n + N} \qquad (12)$$

where $\tau_{\text{min}}$ ($\tau_{\text{maj}}$) is the minority (majority) capture time by deep level defects, $n$ is the excess carrier density, and $N$ is the background doping density[52,75] As increasing of the excess carrier density $n$, Eq. (12) clearly indicates that the $\tau_{SRH}$ would increase, leading to reduction of the corresponding SRH recombination coefficient $A_1$ ($A_1 = 1/\tau_{SRH}$), which is a typical behavior for SRH recombination at an intermediate density regime. On the other hand, the decrease of recombination coefficients $B$ and $C$ with the excess carrier density can be attributed to the phase-space filling effect[63]. It is also interesting to find the exciton recombination coefficient $A_2$ and thus the exciton radiative recombination lifetime $\tau_X$ (i.e., $1/A_2$) are nearly independent with the excess carrier density in range of $7 \times 10^{16}$–$4 \times 10^{17}$ cm⁻³. Such density independent recombination coefficient $A_2$ ($-7.6 \times 10^{10}$ s⁻¹) further evidences its excitonic nature of radiative recombination[76]. Nevertheless, the $A_2$ shows a smaller value ($-3 \times 10^{10}$ s⁻¹) as the photogenerated carrier density decreases to $-10^{16}$ cm⁻³ (with the corresponding exciton density $-10^{14}$ cm⁻³) which might be related

with its more efficient exciton-phonon scattering and weaker exciton-exciton interactions[77].

## Effect of defects on the ultrafast carrier dynamics

As discussed in the previous sections, CuI thin films PTA in $I_2$ and air exhibit very different optoelectronic properties, which is mainly related to differences in defects present in the samples. So far, we only investigated the ultrafast carrier dynamics in the as-grown CuI. It is also interesting to see effects of defects in the PTA CuI thin films on the carrier dynamics. To this end, the TA spectra ($\lambda_p = 320$ nm, $F_p = 2.54$ mJ/cm²) of CuI thin films with different treatments (i.e., as-grown, PTA in air, and PTA in $I_2$) are measured, and compared in Fig. 9a. Generally, the TA spectra for these CuI thin films with different treatments exhibit similar features. The corresponding decay kinetics of the bleach signal at 406 nm as well as their fits are depicted in Fig. 9b. We find that the lifetime of photogenerated carriers increases/decreases upon PTA treatment in $I_2$/air as compared to that of the as-grown CuI thin film. As mentioned earlier, the CuI thin film with PTA in air has higher concentration of deep defects (e.g., $V_I$ and $V_I + Cu_i$ defect clusters) than the as-grown CuI. These deep defects can enhance the overall Auger recombination via the trap-assisted Auger recombination[46,78] resulting in an increased Auger recombination coefficient ($C$-$16.4 \times 10^{-30}$ cm⁶/s). In comparison, the extracted Auger recombination coefficients for the CuI thin film PTA in $I_2$ ($-4.94 \times 10^{-30}$ cm⁶/s) is lower than that of the as-grown

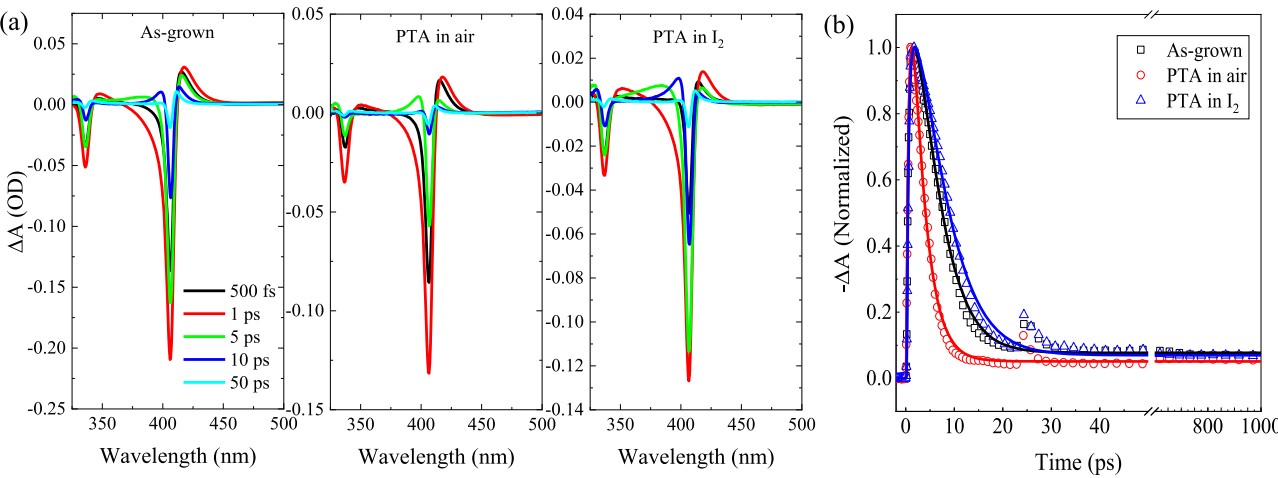

**Fig. 9 | Dependence of ultrafast carrier dynamics on defects. a** TA spectra of CuI thin films with different treatments ($F_p$ = 2.54 mJ/cm², $\lambda_p$ = 320 nm). **b** Normalized PB dynamics probed at the 406 nm, with the lines the corresponding fits.

sample (-7.23 × 10⁻³⁰ cm⁶/s). Therefore, the nonradiative recombination process is aggravated (mitigated) by the increased (decreased) concentration of deep defects in the CuI thin film PTA in air (I₂), leading to shorter (longer) effective carrier lifetime. As mentioned earlier, the radiative recombination in CuI can be mainly attributed to the (e, $V_{Cu}^0$) recombination when the photogenerated carrier density is relatively high (e.g., > 10¹⁹ cm⁻³). The shallow acceptor $V_{Cu}$ can be ionized easily at RT, while these photogenerated free holes can be trapped by these ionized $V_{Cu}$, forming neutral copper vacancies. Thus the (e, $V_{Cu}^0$) recombination can be regarded as a bimolecular recombination process. By using Eq. (9), the bimolecular recombination coefficients $B$ for the CuI thin film after PTA in air and in I₂ are found to be -0.88 × 10⁻¹¹ and -2.06 × 10⁻¹¹ cm³/s, respectively. In comparison with the $B$ for the as-grown sample (i.e., $B$-1.30 × 10⁻¹¹ cm³/s), the radiative recombination coefficient for the sample PTA in air (I₂) decreases (increases), which is reasonable since the conduction band to acceptor transition probability is proportional to the concentration of holes in the acceptor levels (i.e., the concentration of $V_{Cu}^0$ in the case of CuI). This implies that the radiative (e, $V_{Cu}^0$) recombination is more (less) efficient in the CuI thin film after PTA in I₂ (air). Nevertheless, the decreased PL intensity for CuI thin film with PTA in air (see Fig. 3a) is mainly due to its enhanced nonradiative recombination as a consequence of increased deep defects. The bimolecular recombination coefficient $B$ and the Auger recombination coefficient $C$ for the CuI thin films with different treatments are summarized in Supplementary Table 3.

The overall relaxation pathways of photogenerated carriers in the CuI thin films are depicted in the schematic diagrams in Fig. 10. When the excitation density is much higher than the Mott density (in the left panel), ① the generated EHP is thermalized within -0.5 ps via carrier-carrier scattering; ② the EHP with relatively high concentration (>10¹⁹ cm⁻³) undergo a dominating Auger recombination process at a time delay in the range of 1.4–5.5 ps, with the cooling rate of EHP drastically slowed down by the effect of hot-phonon bottleneck as well as the Auger heating, while for the EHP with a medium concentration (10¹⁸–10¹⁹ cm⁻³), the trap-assisted SRH recombination plays the major role in the initial decay process; ③ the radiative bimolecular recombination processes, including the (e, h) recombination and the (e, $V_{Cu}^0$) recombination, occur at a time delay in the range of 30 ps-a few ns to complete the entire recovery process. On the other hand, when the excitation density is less than the Mott density (in the right panel), ① the thermalization of photogenerated carriers is completed at a time delay < 100 fs, and the hot carriers cool rapidly via carrier-phonon scattering with a time constant of

-2 ps; ② the photogenerated carriers undergo a fast trap-assisted SRH recombination at a time delay in the range of 0.75–6 ps, followed by ③ a radiative exciton recombination at a time delay in the range of 6–20 ps; ④ radiative (e, h) recombination and (e, $V_{Cu}^0$) recombination dominate the decay process as the time delay is >20 ps. It is worth noting that in both cases, both EHP and excitons co-exist in the processes of relaxation and recombination.

In conclusion, we comprehensively investigated the optoelectronic properties and the ultrafast carrier dynamics of CuI thin films. The as-grown CuI thin film exhibits hole concentration $N$-9 × 10¹⁹ cm⁻³, and the $N$ decreases (increases) to -3 × 10¹⁹ cm⁻³ (-1.2 × 10²⁰ cm⁻³) after PTA in air (I₂). The free holes can be primarily attributed to shallow native acceptors (e.g., $V_{Cu}$, $I_i$) in the CuI thin films. PTA treatments in air facilitates the formation of native donor defects (e.g., $V_I$, $Cu_i$, $V_I$ + $Cu_i$ defect clusters). The absorption spectra indicate that the excitonic feature of CuI maintains at a high carrier density up to -5 × 10¹⁹ cm⁻³ owing to the possible formation of Mahan excitons at RT. PL measurements reveal that the emission peak at -420 nm is most likely due to the radiative (e, $V_{Cu}^0$) recombination. Femtosecond transient absorption (fs-TA) measurements were carried out to understand the ultrafast carrier dynamics in these CuI thin films. Through carrier-carrier scattering, quasi-equilibrium distribution of photogenerated carriers is rapidly built up (<200 fs) within the time resolution of our TA setup. For relatively high excitation density (>10¹⁹ cm⁻³), the EHP underwent a fast Auger recombination (with Auger recombination coefficient of -7.42 × 10⁻³⁰ cm⁶/s) in the initial decay process with a time delay < 5–6 ps, while the carrier cooling rate (with cooling time -6 ns) slows down notably due to the effects of hot-phonon bottleneck as well as the Auger heating. Interestingly, a re-excitation was also found at a time delay -25 ps, which is most probably due to the stimulated emission from the degenerate EHP. As in the case of medium excitation density (10¹⁸–10¹⁹ cm⁻³), the photogenerated carriers rapidly cool down with a cooling time < 3 ps, and the initial recombination is governed by the trap-assisted SRH process (with the SRH recombination coefficient of -2.42 × 10¹¹ s⁻¹). When the excitation density is relatively low (<10¹⁸ cm⁻³), the SRH recombination plays the major role in the initial decay process with a time delay < 6 ps, followed by a radiative exciton recombination (with recombination coefficient of -6.4 × 10¹⁰ s⁻¹) at a time delay in the range of 6–20 ps. As the time delay is >20 ps, the recombination kinetics is dominated by the radiative bimolecular recombination process (with recombination coefficient of -1.3 × 10⁻¹¹ cm³ s⁻¹). As increasing of excess carrier density, recombination coefficients linked to the SRH recombination, bimolecular recombination and Auger recombination decrease, while the radiative

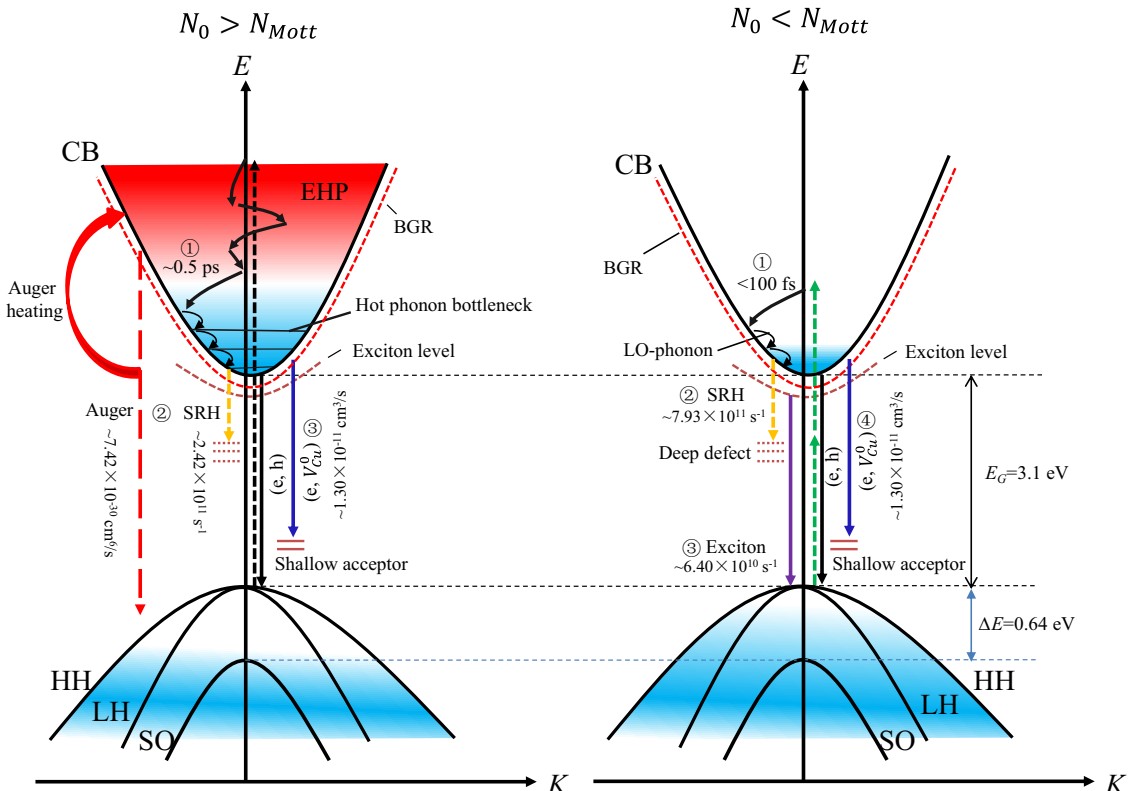

**Fig. 10 | Schematic illustration of carrier dynamics in CuI thin film.** The left panel and the right panel represent the cases with excitation density ($N_0$) larger and less than the Mott density ($N_{Mott}$), respectively.

exciton recombination coefficient is nearly constant. Two-photon absorption in the CuI thin films was verified from the fs-TA measurements as well as the Z-scan measurements. Two-photon induced ultrafast carrier dynamics was also discussed. CuI thin films with different PTA treatments show distinct defect-related recombination behaviors, i.e., the nonradiative recombination in the CuI thin film PTA in air is accelerated by deep defects via the trap-assisted Auger recombination, while the radiative (e, $V_{Cu}^0$) recombination is more efficient in the CuI thin film TPA in $I_2$ due to its higher density of $V_{Cu}^0$.

## Methods

### Synthesis of CuI thin films

The precursor $Cu_3N$ thin films were deposited on glass and quartz substrates by reactive radio-frequency (RF) sputtering of a metallic Cu target in a mixed atmosphere of $N_2$ and Ar (flow ratio of $f(N_2):f(Ar) = 2:1$) without intentional substrate heating. The CuI thin films with thickness ~96 nm were subsequently obtained by solid iodization of the $Cu_3N$ precursor films with sufficient iodine powder to cover the $Cu_3N$ thin films in a sealed container at room temperature (RT) for around 20 min. Post-growth thermal annealing (PTA) at 100 °C for 10 min was carried out either in air or iodine ($I_2$) vapor to further control the defect concentration.

### Film characterizations

The crystal structure was analyzed by using grazing incidence x-ray diffraction (GIXRD) with a grazing incidence angle of 1°. Raman spectra were measured at RT under the excitation of a 532 nm-wavelength laser by using a Jobin Yvon LabRAM HR 800 UV micro-Raman system. The surface morphology and surface roughness were characterized by using field emission scanning electron microscopy (SEM; Gemini 300, Carl-Zeiss) and the atomic force microscopy (AFM; Dimension Icon, Bruke), respectively. Optical properties of CuI thin films were obtained from standard spectroscopic ellipsometry (SE) measurements performed at

RT in the photon energy range of 0.75–5.9 eV at incidence angles of $\Phi = 65°$ using a commercial rotating-compensator ellipsometer (Eoptics, SE-VM-L) in the polarizer-compensator-sample-compensator-analyzer (PCSCA) configuration. Electrical properties were investigated by Hall-effect measurements in the van der Pauw configuration. Steady-state PL spectra were recorded under the excitation wavelength of 320 nm using a xenon lamp as the excitation source. Pump-probe fs-TA spectroscopic measurements were performed at RT in the spectral range of 320–700 nm using a ~84 fs pump pulse with varying pump fluence ($F_p$) and pump wavelength ($\lambda_p$). The excitation source was an 800 nm pulsed laser with a pulsed duration of 84 fs and a repetition rate of 1 kHz. The pump laser with different wavelengths were produced by using an optical parametric amplifier (OPA, TOPAS-C, Light Conversion) system, while a broadband (320–820 nm) white-light probe beam was generated via supercontinuum generation from a 2 mm thick sapphire or $CaF_2$ plate by focusing the 1 kHz Ti:Sapphire amplified laser (Coherent, Astrella-Tunable-F-1k). The probe beam is vertical to the sample, with pump-probe angle of ~5°. The same laser source was used to study the nonlinear absorption using the closed-aperture Z-scan technique.

## Data availability

The source data that support the findings of this paper are provided with this paper. Source data are provided with this paper.

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

## Acknowledgements

Z.H.L., X.H.L., and C.P.L. acknowledge financial support from the Department of Science and Technology of Guangdong Province under Project No. 2021A0505030081, Guangdong Basic and Applied Basic Research Foundation (Project No. 2020A1515010180), Guangdong University Key Platform (Grant No. 2021GCZX009), the start-up fund from Shantou University under Project No. NTF18027. K.M.Y. acknowledges financial support of the CityU SGP (Grant No. 9380076).

## Author contributions

Z.H.L. performed the experiments, analyzed the data, and wrote the original draft. J.X.H. helped the ultrafast spectroscopy measurerments. X.H.L., L.F.C., and K.O.E. helped the thin film characterizations. C.P.L. conceived the original concept and supervised the project. Z.H.L., M.D.L., T.T., Q.X.G., K.M.Y., and C.P.L. revised the manuscript.

## Competing interests

The authors declare no competing interests.

## Additional information

**Correspondence and requests** for materials should be addressed to Ming-De Li or Chao Ping Liu.

