## [Peer Review File · Nature Communications]

Title: Optoelectronic properties and ultrafast carrier dynamics of copper iodide thin filmsREVIEWER COMMENTS

Reviewer #1 (Remarks to the Author):

The manuscript by Li et al. provides a comprehensive overview of the optoelectronic properties, starting with the steady-state properties and the effects of post-growth sample treatment, followed by a detailed discussion of the carrier dynamics after pulsed laser excitation. Thus, the paper contains a wealth of information/discussion, some of which is illuminating, if not entirely conclusive. I believe that the manuscript is valuable to a wide readership and deserves publication if the authors can provide some clearer explanations and make a number of corrections.

The specific comments are listed below:

The authors mention that the grown CuI thin films have a thickness of about 96 nm, which is quite comparable to the calculated grain size value. What are the variations/scattering of the film thickness for different grown CuI thin films? Could the FWHM simply be limited by the film thickness? Could the authors also comment on any possible changes of the film thickness after PTA procedure?

The decreased FWHM of the TO mode in the Raman spectra for the sample baked in air was attributed to a larger grain size after PTA in air. However, the grain size is still larger for the sample after PTA in I₂, although the TO-FWHM is not significantly different from that of the as-grown film. This also makes the correlation between the grain size and FWHM for PTA in air unconvincing.

The effects of PTA on the electrical properties and the decrease in resistivity after treatment in air are explained. This behavior is attributed to the annihilation of V_{Cu} and the out-diffusion of I_i. However, oxygen has been shown to be a fairly shallow acceptor leading to an increase in carrier density even when the sample is exposed to air/oxygen at room temperature (Storm et al., APL Materials 8, 091115 (2020)). In addition, there is also the possibility of oxidation of Cu to CuO (Lin et al., Materials 9, 990 (2016)). Can the authors say anything about the incorporation of oxygen into the thin films after PTA?

The steady-state optical properties of the CuI thin films were investigated using SE at room temperature. The authors should describe the modeling of the raw data in some more detail. Was a numerical approach or an analytical model used to model the optical properties of CuI? How well does the model describe the experimental data? The authors mention that SE was measured in the spectral range between 0.79 eV and 5.9 eV. However, no data are given for the absorption coefficient below 1.5 eV. Is there any evidence for absorption by free charge carriers in the NIR, which is to be expected due to the high charge carrier density (see e.g. Krüger et al., Appl. Phys. Lett. 113, 172102 (2018))?

What is the reason for the increased absorption between the Z_{1,2} and Z₃ resonances of the PTA in the I₂ sample compared to the as-grown and air-annealed samples?

In addition, it should be noted that the yellowish coloration of the thin film sample is not necessarily evidence of I₂ adsorption on the film surface, but may be caused by the absorption coefficient of the

material and the modulation of the reflectance/transmittance due to thickness ("Fabry-Perot") oscillations.

The shift of the main PL emission peak by about 21 meV to lower energies with aging was attributed to BGR caused by the decreasing carrier density. However, the carrier density changes by only 40% within the first 10 days. Such an effect could not be clearly established from the measured absorption coefficients at steady state for the untreated and the PTA samples, where the differences in carrier density are comparable. Furthermore, since the emission profile may consist of an overlap of different transitions, it seems hard to make a clear statement about the energy shift of the band gap with aging time.

The main part of the manuscript deals with the discussion of transient absorption (TA) spectra. It is well known that photo-excited states lead to changes in both the absorption and reflectivity of the material (Richter et al., *New Journal of Physics* 22, 083066 (2020)). For low refractive index materials, the change in reflectivity can be neglected to a good approximation, so that transient absorption is often determined directly from transient transmission. However, for materials with relatively high refractive index the change in reflectivity may need to be taken into account data, as was shown e.g. for lead-halide perovskites (Price et al., *Nature Communications* 6, 1 (2015)). Therefore, the possible influence of the refractive index change on the TA data should also be discussed for CuI. In addition, the photoluminescence may also be able to be detected along with the transmittance signal, since the sample thickness is comparable to the penetration depth of the pump laser and thus the sample is excited almost homogeneously. A detailed description of the experimental technique and the corrections used by the authors (e.g., for reflectivity or PL) is therefore essential for understanding (or the ability to reproduce) the results presented and should be included in the manuscript.

The authors discuss that the optically generated charge carrier density is one to two orders of magnitude larger than the calculated charge carrier density at the Mott transition, so that excitonic effects can be neglected. Recently, however, it has been shown that in ZnO, for example, the excitonic features are still present in the dielectric function even around the Mott transition, indicating that the occupation of the exciton ground state does not exceed the Mott density even at high total carrier densities (Richter et al., *New Journal of Physics* 22, 083066 (2020)). Moreover, it has been observed that electron-hole coupling is maintained even above the Mott transition, leading to so-called Mahan excitons (Mahan, *Physical Review* 153, 882 (1967) & Zimmermann, *physica status solidi (b)* 146, 371 (1988)).

Therefore, I consider the exclusion of excitonic effects based on the calculated charge carrier density alone to be premature. In the grown samples, for example, the charge carrier density already exceeds the calculated Mott charge carrier density by an order of magnitude. Nevertheless, a clear excitonic response can be seen in the absorption spectra, indicating that excitonic effects must be expected even at high charge carrier densities. Is there any evidence of complete bleaching of the excitonic resonance in the transient transmission spectra? A selection of time-delayed transient transmission spectra seems essential and should be added, also to understand the BGR discussed later as well as the photo-induced

absorption below the band gap.

The increase in ΔA at a time delay of about 25 ps is attributed to the re-excitation of charge carriers by stimulated emission. However, the detection of photoluminescence would also lead to an effective increase (decrease) in transmission density (absorption) at the emission wavelength. Moreover, the discussion is based on the statement that stimulated emission was observed in CuI microwires 30 ps after optical excitation. However, Wille et al. (Applied Physics Letters 111, 031105 (2017)) show that PL emission starts almost immediately after the excitation pulse hits the sample.

Due to high laser fluencies, the material can be locally strongly heated, as also mentioned by the authors. It is conceivable that such heating of the material can lead to damage and even oxidation of the CuI surface in air. Are the initial absorption spectra ever fully recovered or are the changes in optical properties irreversible after pump?

At a pump fluence of $> 1.5 \text{ mJ/cm}^2$, the maximum signal ΔA saturates to a constant value, possibly indicating that the excitonic absorption is indeed fully bleached at the corresponding carrier densities and that the Mott transition is thus reached here. Could the authors comment on this?

The increased absorption above the band gap is attributed to the widening of the CuI band gap with increasing temperature predicted by Xu et al. [1]. However, an increase in the band gap in CuI with increasing temperature has not been experimentally demonstrated, at least up to 300 K [2-4]. In addition, it should be noted that similar behavior above the band gap has been observed in lead halide perovskites [5], but where it was attributed only to changes in reflectivity rather than a change in absorption.

[1] Xu et al., Journal of Physics: Condensed Matter 34, 134002 (2022)

[2] Serrano et al., Physical Review B 65, 125110 (2002)

[3] Krüger et al., Appl. Phys. Lett. 113, 172102 (2018)

[4] Krüger et al., APL Materials 9, 121102 (2021)

[5] Price et al., Nature Communications 6, 1 (2015)

For the case of two-photon induced charge carrier dynamics, the charge carrier densities were estimated to be between 4×10^{17} and 8×10^{18} . Could the author clarify how exactly the charge carrier density was calculated for two-photon excitation?

Reviewer #2 (Remarks to the Author):

Optoelectronic properties and ultrafast carrier dynamics of copper iodide thin films

The proposed work puts forward a study of copper iodide (CuI) as an optoelectronic material. The study is motivated by growing interest on transparent conductors, and the potential of CuI as a p-type transparent conductor. CuI benefits from a mobility that is higher than comparable p-type transparent conductors, although its mobility is still relatively low for envisioned/future applications. The mobility is known to be affected by defect states, amongst other mechanisms, and the proposed work studies the underlying mechanisms.

The proposed work puts forward rigorous analyses of CuI. The analyses include x-ray diffraction and Raman spectroscopy, to characterize the crystallinity, electrical measurements, to characterize the effects of growth conditions and time on the mobility, and pump-probe/transient-absorption analyses, to characterize the ultrafast relaxation in the material. The authors are to be commended for this breadth of analysis. It can help resolve ongoing debates in literature on the relaxation pathways for charge carriers and excitons in CuI. The authors' characterizations of trends in photoluminescence intensity and recombination pathways versus charge carrier density were especially insightful for this. They were useful in distinguishing the combined effects of trap-assisted, bimolecular, and Auger recombination. However, a few more details should be explained or addressed in the use of equation (3). This equation plays a central role in the analysis, as it was used to estimate the excited charge carrier density, but its validity depends upon the penetration depth of the pump beam, i.e., the reciprocal of the absorption coefficient. A large penetration depth, in comparison to the thickness of the film, could yield a substantial reflection from the back surface of the film with a corresponding enhancement to the charge carrier density. Further information is needed here for the pump wavelength, its penetration depth, and the validity of equation (3). Other than this, the authors put forward a rigorous characterization of CuI and make convincing arguments for the charge carrier dynamics within this material.

Reviewers' comments:

Reviewer #1 (Remarks to the Author):

The manuscript by Li et al. provides a comprehensive overview of the optoelectronic properties, starting with the steady-state properties and the effects of post-growth sample treatment, followed by a detailed discussion of the carrier dynamics after pulsed laser excitation. Thus, the paper contains a wealth of information/discussion, some of which is illuminating, if not entirely conclusive. I believe that the manuscript is valuable to a wide readership and deserves publication if the authors can provide some clearer explanations and make a number of corrections.

The specific comments are listed below:

Response:

We thank the reviewer for his/her positive evaluation of our manuscript. In the following, we will answer these questions raised.

1. The authors mention that the grown CuI thin films have a thickness of about 96 nm, which is quite comparable to the calculated grain size value. What are the variations/scattering of the film thickness for different grown CuI thin films? Could the FWHM simply be limited by the film thickness? Could the authors also comment on any possible changes of the film thickness after PTA procedure?

Response:

The grain sizes reported in the original manuscript were inaccurate due to the failure in convergence for the fitting. We recalculated the grain size for the CuI thin films by the Scherrer equation using the width of the (111) diffraction peak. We now estimate the grain sizes to be 30.5 nm (FWHM=0.264°), 32.5 nm (FWHM=0.248°) and 37.1 nm (FWHM= 0.217°) for the as-grown CuI, CuI with PTA in I₂, and CuI with PTA in air, respectively. We apologize for such mistake. From our SE analysis, the thickness of CuI thin film with PTA in air or I₂ is very close to the corresponding as-grown CuI thin film, with variation in thickness less than 2 nm. We further compared the FWHM of XRD peaks for CuI thin films with different thicknesses. For the as-grown CuI thin films, the 2θ FWHM is found to decrease from 0.308° to 0.119° as increasing of thickness from ~60 nm to ~300 nm (see Supplementary Fig. 1). The FWHM would further decrease upon annealing treatments. Accordingly, we revised corresponding description on page 3 in the revised manuscript.

“The grain sizes were calculated by the Scherrer equation from the widths of the(111) diffraction peak. For the CuI with thickness ~100 nm, the grain size (D) for films with PTA in air (D~37.1 nm) and I₂ (D~32.5 nm) is larger than that of the as-grown (D ~30.5 nm) sample. Apparently, such annealing treatments in air or I₂ can enhance the crystallinity of CuI thin film. In order to investigate the dependence of FWHM of XRD peak or the grain size on the film thickness, we further carried out XRD measurements for the as-grown CuI thin films with different thicknesses. The FWHM of the (111)

diffraction peak is found to decrease from 0.308° to 0.119° , corresponding to grain size increasing from 26.1 to 67.7 nm, as the film thickness increases from ~60 nm to ~300 nm (see Supplementary Fig. 1). Therefore, the crystallinity of as-grown CuI thin film improves with film thickness. In the following sections, results were taken from films with thickness of ~100 nm, unless otherwise specified.”

2. The decreased FWHM of the TO mode in the Raman spectra for the sample baked in air was attributed to a larger grain size after PTA in air. However, the grain size is still larger for the sample after PTA in I₂, although the TO-FWHM is not significantly different from that of the as-grown film. This also makes the correlation between the grain size and FWHM for PTA in air unconvincing.

Response:

Actually, both the grain size and the concentration of defects can significantly affect the broadening of Raman peaks of materials (see Gouadec et al., Raman spectroscopy of nanomaterials: How spectra related to disorder, particle size and mechanical properties, Progress in Crystal Growth and Characterization of Materials 53, 1, 2007). Therefore, the Raman peak width in our CuI thin film is mainly determined by its grain size as well as the concentration of defects involved. In order to further unravel effect of grain size and the defects on the Raman peak width of CuI thin film, we annealed the as-grown thin films at varying annealing temperature (100 ~300 °C) in air for 10 minutes. As seen in Fig. R1(a), the FWHM of XRD for these samples decrease from 0.152 to 0.108 (with the grain size increasing from 52 nm to 74 nm), while the corresponding FWHM of Raman peak (see Fig. R1(b)) decreases from 16 to 14 cm⁻¹. The relatively larger FWHM of Raman peak as observed in the CuI thin film with PTA in I₂ is most likely due to its much higher concentration of acceptor defects (e.g., iodine interstitials, copper vacancies, etc). Accordingly, we revised corresponding description on page 3 in the revised manuscript.

“In contrast, the relatively wide TO peak observed for the CuI films as-grown and PTA in I₂ atmosphere may be due to their relatively high concentration of acceptor defects (e.g., iodine interstitials I_i, V_{Cu} etc).”

Fig. R1. XRD patterns (a) and the corresponding Raman spectra (b) for CuI thin film with PTA treatments in air at different temperatures.

3. The effects of PTA on the electrical properties and the decrease in resistivity after treatment in air are explained. This behavior is attributed to the annihilation of V_{Cu} and the out-diffusion of I_i . However, oxygen has been shown to be a fairly shallow acceptor leading to an increase in carrier density even when the sample is exposed to air/oxygen at room temperature (Storm et al., APL Materials 8, 091115 (2020)). In addition, there is also the possibility of oxidation of Cu to CuO (Lin et al., Materials 9, 990 (2016)). Can the authors say anything about the incorporation of oxygen into the thin films after PTA?

Response: Indeed, there are some reports about the p-type conductivity in CuI arising from the O involved by means of possible oxidation (e.g., Storm et al., APL Materials 8, 091115 (2020)). However, in our experiments, the hole concentration of grown CuI thin film did not show any noticeable enhancement upon aging in ambient air. Based on DFT calculations, Grauzinyte et al. (Phys. Chem. Chem. Phys 21, 18839, 2019) found that the substitution of iodine by oxygen (i.e., O_I) have a rather low equilibrium concentration ($\sim 10^6 \text{ cm}^{-3}$) at room temperature due to its high formation energy. In the work reported by Lin et al. (Materials 9, 990 (2016)), the CuO phase (evidenced by the XRD) was found in the CuI thin film upon post thermal annealing in air at high temperature (i.e., 300 °C) for 2 hours. In comparison, we annealed the samples at a much milder condition (e.g., low temperature of 100 °C and short duration of 10 min), and no CuO phase was discernable from the XRD spectra. Therefore, we believe that the increase in hole concentration in our CuI thin films can be mainly attributed to their native defects (e.g., V_{Cu} and or I_i) rather than the possible incorporation of O. Accordingly, we clarify this on page 5 in the revised manuscript.

“Regarding the high N in the nominally undoped CuI, possible contributions from the oxidation and incorporation of atmospheric oxygen into the CuI thin film were proposed recently by Storm et al[35]. However, in our experiments, the N of CuI thin

films did not show any noticeable enhancement upon aging in ambient air. Based on DFT calculations, Grauzinyte et al[25]. found that due to its high formation energy, the substitutional oxygen in the iodine sublattice (i.e., O_I) have a rather low equilibrium defect concentration ($\sim 10^6 \text{ cm}^{-3}$) at room temperature. Therefore, we believe that the hole concentration in our CuI thin films is mainly attributed to their native defects (e.g., V_{Cu} and or I_i) rather than the possible incorporation of O”

4. The steady-state optical properties of the CuI thin films were investigated using SE at room temperature. The authors should describe the modeling of the raw data in some more detail. Was a numerical approach or an analytical model used to model the optical properties of CuI? How well does the model describe the experimental data? The authors mention that SE was measured in the spectral range between 0.79 eV and 5.9 eV. However, no data are given for the absorption coefficient below 1.5 eV. Is there any evidence for absorption by free charge carriers in the NIR, which is to be expected due to the high charge carrier density (see e.g. Krüger et al., Appl. Phys. Lett. 113, 172102 (2018))?

Response: SE analysis was based on a three-layer optical model, i.e., the glass substrates, CuI thin film, and a surface roughness layer. For CuI thin films, a general Herzinger Johs Parameterized Semiconductor Oscillator (Psemi-M0), a Tauc-Lorentz oscillator, and three Gaussian oscillators and a Drude oscillator were used to describe their dielectric functions. These dielectric function models can fit the experimental SE data (Ψ , Δ) quite well over the whole spectral range (see Fig.3(a,b) in the revised manuscript), with a mean square error (MSE) around 2.7. Be noted that index grading was used for the CuI layer in the optical model. In the revised manuscript, the absorption coefficient spectra in the low energy range is also shown in Fig. 3(c). Distinct free carrier absorption is presented for the CuI thin films with relatively high hole densities (i.e., as-grown, PTA in I_2). Accordingly, we added/changed corresponding description on page 6 in the revised manuscript.

“SE analysis was based on a three-layer optical model, i.e., the glass substrates, CuI thin film, and a surface roughness layer. For CuI thin films, a Psemi-M0 oscillator, a Tauc-Lorentz oscillator, three Gaussian oscillators and a Drude oscillator were used to describe their dielectric functions. These dielectric function models can fit the experimental SE data (Ψ , Δ) quite well over the whole spectral range, with a mean square error (MSE) around 2.7. Fig. 3(a) and Fig. 3(b) show the measured and the fitted Ψ and Δ , respectively for the as-grown CuI thin film with the angle of incidence of 65° . Be noted that index grading was used for the CuI layer in the optical model. The absorption coefficient (α , obtained from SE analysis) of the CuI thin film with different treatments are shown in Fig. 3(c). The excitonic band edge absorption at $E_0 \sim 3.06 \text{ eV}$ (denoted as $Z_{1,2}$) and the absorption peak at $(E_0 + \Delta_0) \sim 3.7 \text{ eV}$ (denoted as Z_3) are distinct for the CuI thin film with PTA in air, while they both almost disappear for the as-grown and the CuI thin film with PTA in I_2 . This suggests that the excitonic effects in the CuI thin film are negligible as the carrier density is larger than $\sim 8 \times 10^{19} \text{ cm}^{-3}$, due to the

screening of Coulomb interaction. The free carrier absorption (at photon energies <1.2 eV) is notable for the CuI thin films with relatively high hole density ($>8 \times 10^{19} \text{ cm}^{-3}$), as seen from Fig. 3(c). In comparison with the as-grown CuI thin film, the CuI thin film with PTA in I₂ shows negligible change in α , while the CuI thin film with PTA in air shows noticeable reduction in sub-gap absorption as well as the free carrier absorption (<1 eV). The origin of the sub-gap absorption is related with native defects (e.g., V_{Cu} , V_{I} and $V_{\text{I}+\text{Cu}}$ defect clusters, etc) and/or the defects at grain boundaries.”

5. What is the reason for the increased absorption between the Z_{1,2} and Z₃ resonances of the PTA in the I₂ sample compared to the as-grown and air-annealed samples?

Response: We apologize for a mistake we made in the previous manuscript. At the early stage of this research work, we needed to send the samples to CityU for the SE measurements, which inevitably delayed the measurements. The absorption coefficient for these CuI thin films were actually for samples with quite long aging time (~>60 days), and the hole densities in these aged samples likely reduced significantly. In the revised manuscript, we synthesized new CuI thin films (as-grown, PTA in I₂, PTA in air, etc) and carried out the SE measurements for these fresh prepared samples (now we have the ellipsometer in Shantou University). As seen from the Fig. 3(c) in the revised manuscript, the absorption coefficient α between the Z_{1,2} and Z₃ resonances is slightly dependent on their carrier densities N , i.e., sample with higher N exhibits higher α but with lower α at energies close to the Z_{1,2} resonance.

6. In addition, it should be noted that the yellowish coloration of the thin film sample is not necessarily evidence of I₂ adsorption on the film surface, but may be caused by the absorption coefficient of the material and the modulation of the reflectance/transmittance due to thickness ("Fabry-Perot") oscillations.

Response: In our experiment, we found that such yellowish color would disappear after several days. Also, CuI thin films with different thicknesses can show such yellowish color after PTA in I₂. Therefore, we believe that the yellowish coloration of the CuI thin film after PTA in I₂ is mainly due to the excessive I₂ adsorption on the film surface and at the grain boundaries. We also note that the yellowish coloration only appears on CuI thin film after PTA treatment in I₂ with relatively high concentration of I₂ vapor. Nevertheless, the exact origin of color in CuI is still quite controversial. For example, the adsorbed iodine species on the surface of CuI was ascribed to its color [P. Gao, et al. Cryst. Eng. Comm. 15, 2934 (2013)], while possible defects (e.g., V_{Cu} , V_{I} , and I_{Cu}) in the bulk crystal rather than the surface states was believed to account for its unique optical properties (Koyasu et al., J. Appl. Phys. 125, 115101, 2019). Since native defects in CuI are unstable, the yellowish color on the fresh CuI thin film with PTA in I₂ may also be related to the evolution of native defects involved. Accordingly, we added corresponding description on page 6 in the revised manuscript.

“We also note that excessive I₂ adsorption on the film surface and at the grain boundaries may be present in the case of relatively high concentration of I₂ vapor in the PTA treatment[39], which was verified by its slightly yellowish color as well as its high sub-gap absorption coefficient (not shown). Nevertheless, the exact origin of color in CuI is still quite controversial. For example, Gao et al. ascribed this coloration of CuI to adsorbed iodine species on the crystal surface[40], while Koyasu et al. suggested that possible native defects (e.g., V_{Cu}, V_I, and I_{Cu}) in the bulk crystal rather than the surface states may account for its unique optical properties[41]. Since native defects in CuI are rather unstable, the yellowish color from our fresh CuI thin film with PTA in I₂ may also be related to the evolution of native defects involved.”

7. The shift of the main PL emission peak by about 21 meV to lower energies with aging was attributed to BGR caused by the decreasing carrier density. However, the carrier density changes by only 40% within the first 10 days. Such an effect could not be clearly established from the measured absorption coefficients at steady state for the untreated and the PTA samples, where the differences in carrier density are comparable. Furthermore, since the emission profile may consist of an overlap of different transitions, it seems hard to make a clear statement about the energy shift of the band gap with aging time.

Response:

In order to further investigate the possible BGR effect in CuI thin films, the optical properties of CuI thin films with hole densities ranged from 4.16×10^{19} to $1.68 \times 10^{20} \text{ cm}^{-3}$ (achieved by PTA in air at 100 °C for varying durations) were studied by using SE.

The optical band gap E_G^{opt} was obtained by extrapolating α^2 to zero photon energy.

With increasing hole density, the optical band gap (E_G^{opt}) of CuI thin film increases

from ~2.97 eV to ~2.972 eV. The optical band gaps obtained for samples correspond to their absorption edges. Furthermore, we calculated the energy difference between the Fermi level E_F and the valence band edge E_V (i.e., $E_F - E_V$) by using $N =$

$2 \frac{m_h k T^{3/2}}{2\pi \hbar^2} F_{1/2}\left(-\frac{E_F - E_V}{kT}\right)$, where N is the hole density in the valence band, m_h is the

density of state effective hole mass ($m_h = (m_{hh}^{3/2} + m_{lh}^{3/2})^{2/3}$) which is found to be

around $2.47 m_e$ for CuI (Grundmann et al., Phys. Status Solidi A 210, 1671, 2013), k is

the Boltzmann constant, T is the temperature, \hbar is the Plank constant, $F_{1/2}$ is the

Fermi-Dirac integral which was calculated by the method proposed by Bednarczyk et

al. (Phys. Lett. 64A, 409, 1978). The obtained result is given in Fig. R2. It is found that

$E_V - E_F$ is around 0, and 29 meV for samples with hole density $N \sim 7.6 \times 10^{19} \text{ cm}^{-3}$,

$1.68 \times 10^{20} \text{ cm}^{-3}$, respectively. The Burstein-Moss shift due to the hole filling in the

valence band of CuI thin film is approximately equal to the $E_V - E_F$, i.e., $\Delta E_{BM} = E_V - E_F$.

The optical band gap can be expressed as $E_G^{opt} = E_G + \Delta E_{BM} - \Delta E_{BGR}$, in which we took $E_G=2.97$ eV and $\Delta E_{BGR}=0$ for sample with $N=4.16 \times 10^{19} \text{ cm}^{-3}$ as reference. The ΔE_{BGR} for CuI thin film with $N \sim 1.68 \times 10^{20} \text{ cm}^{-3}$ is calculated to be 27 meV, so that the optical gap is nearly unchanged due to the offset by the ΔE_{BM} . It is interesting to find that this ΔE_{BGR} is quite close to the energy shift (i.e., 21 meV) as observed from the PL spectra. It is worth mentioning that optical absorption edge is related with the effects of BGR and BM shift, while PL peak is not affected by the BM shift. We also note that there is no ΔE_{BGR} when E_F is above the E_V or when the hole densities is relatively low (e.g., $< 8 \times 10^{19} \text{ cm}^{-3}$). In which case, the obtained E_G^{opt} corresponds to the excitonic absorption edge. Therefore, the observed PL peak shift with aging time is very likely due to the BGR effect. Nevertheless, the band tails of the CuI thin film and/or the broadening of the defect (V_{Cu}) band may also change upon variation in corresponding defect density. Thus, the PL peak shift with aging time may be attributed to the BGR effect and/or the variations in the band tails/broadening of the defect band in the CuI thin film. Accordingly, we changed corresponding description on page 7~8 in the revised manuscript.

“In order to better understand this PL peak shift, the optical properties of CuI thin films with N ranged from 4.16×10^{19} to $1.68 \times 10^{20} \text{ cm}^{-3}$ (achieved by PTA in air at 100 °C for varying durations) were studied by using SE. The optical band gap E_G^{opt} was obtained by extrapolating α^2 to zero photon energy. As the hole density increases, the E_G^{opt} of CuI thin film increases from ~ 2.97 eV to ~ 2.972 eV. It is worth noting that E_G^{opt} obtained for CuI thin films with distinct excitonic absorption (i.e., with $N < 5 \times 10^{19} \text{ cm}^{-3}$) correspond to their excitonic absorption edges which is less than its exciton peak energies (~ 3.06 eV). The energy difference between the valence band edge E_V and the Fermi level E_F (i.e., $E_V - E_F$) was calculated by using $N = 2 \frac{m_h kT^{3/2}}{2\pi\hbar^2} F_{1/2}(-\frac{E_V - E_F}{kT})$, where N is the hole density in the valence band, m_h is the density of state hole effective mass ($m_h = (m_{hh}^{3/2} + m_{lh}^{3/2})^{2/3}$) which is found to be around $2.47 m_e$ for CuI [16], k is the Boltzmann constant, T is the temperature, \hbar is the Plank constant, $F_{1/2}$ is the Fermi-Dirac integral which was calculated by method proposed by Bednarczyk et al.[45]. It is found that $E_V - E_F$ is around 0, and 29 meV for samples with hole density $N \sim 7.6 \times 10^{19}$ and $1.68 \times 10^{20} \text{ cm}^{-3}$, respectively. The Burstein-Moss (BM) shift (ΔE_{BM}) due to the hole filling in the valence band of CuI thin film is approximately equal to the $E_V - E_F$, i.e., $\Delta E_{BM} = E_V - E_F$. The optical band gap can be expressed as $E_G^{opt} = E_G + \Delta E_{BM} - \Delta E_{BGR}$, where ΔE_{BGR} is the band gap narrowing due to the band gap renormalization (BGR), and we took $E_G=2.97$ eV and $\Delta E_{BGR}=0$ for sample with

$N=4.16 \times 10^{19} \text{ cm}^{-3}$ as reference. It is interesting to find that the ΔE_{BGR} for CuI thin film with $N \sim 1.68 \times 10^{20} \text{ cm}^{-3}$ is $\sim 27 \text{ meV}$, which is quite close to the energy shift (i.e., 21 meV) as observed in the PL spectra (Fig. 4b). It is worth mentioning that optical absorption edge is related with the effects of BGR and BM shift, while PL peak is not affected by the BM shift. However, we also noted that there is no ΔE_{BGR} for CuI thin film with relatively low hole densities (e.g., $< 8 \times 10^{19} \text{ cm}^{-3}$), whose E_F is above the corresponding E_V (i.e., the obtained E_G^{opt} corresponds to the excitonic absorption edge). Therefore, the observed PL peak shift with aging time is very likely due to the BGR effect. Nevertheless, upon variation in concentration of defect (e.g., V_{Cu}) densities, the band tails of CuI thin film and/or the broadening of the corresponding defect band may also change and thus cause the PL peak shift. Hence, the PL peak shift with aging time as observed in Fig.4(b) may be attributed to the BGR effect and/or the variations in the band tails/broadening of the defect band in the CuI thin film[47]”

Fig. R2. Calculated energy Fermi level positions with respect to the valence band maximum ($E_V - E_F$) as function of hole density for CuI thin films.

8. The main part of the manuscript deals with the discussion of transient absorption (TA) spectra. It is well known that photo-excited states lead to changes in both the absorption and reflectivity of the material (Richter et al., New Journal of Physics 22, 083066 (2020)). For low refractive index materials, the change in reflectivity can be neglected to a good approximation, so that transient absorption is often determined directly from transient transmission. However, for materials with relatively high refractive index the change in reflectivity may need to be taken into account data, as was shown e.g. for lead-halide perovskites (Price et al., Nature Communications 6, 1

(2015)). Therefore, the possible influence of the refractive index change on the TA data should also be discussed for CuI. In addition, the photoluminescence may also be able to be detected along with the transmittance signal, since the sample thickness is comparable to the penetration depth of the pump laser and thus the sample is excited almost homogeneously. A detailed description of the experimental technique and the corrections used by the authors (e.g., for reflectivity or PL) is therefore essential for understanding (or the ability to reproduce) the results presented and should be included in the manuscript.

Response:

Indeed, the transient reflectance for samples with relatively high refractive index may need to be taken into account for accurate determination of transient absorption. For such investigation, the incident angles of probe beams should be identical for the transient transmission and reflectivity measurements. To this end, we performed the transient transmission and reflection spectroscopy measurements with probe incident angle of 45° (with schematic diagram of such measurements shown in Supplementary Fig. 4). It is shown that neglecting the reflection change ΔR would not qualitatively change the results. As for the transient PL measurements, unfortunately, we are unable to carry out such measurement by using our present ultrafast spectroscopy system. Accordingly, we added corresponding description on page 13 in the revised manuscript.

“It is worth mentioning that the accurate determination of transient absorption may also need to take into account the transient reflectance as well, though the change in reflectivity can be neglected for low refractive index materials. In order to verify the effect of reflectivity change on the overall absorption change, we performed the transient transmission and transient reflection measurements with incident angle of probe beam of 45° (see the Supplementary Fig. 4 for the schematic diagram). By only taking into account the reflection at the air/CuI interface, the transient absorption

coefficient change $\Delta\alpha$ can be expressed as, $(-\Delta\alpha)l = \frac{\Delta T}{T} + \frac{R}{1-R} \frac{\Delta R}{R}$, where l is the

path length of the transmitted light in the CuI thin film, $\frac{\Delta T}{T}$ is the relative transmission

change, $\frac{\Delta R}{R}$ is the relative reflection change, R is the reflectance (incident angle of 45

°) at the steady-state (i.e., without pump excitation) which can be derived from the SE analysis. Supplementary Fig. 5(a) shows the relative transmission change $\Delta T/T$, relative reflectivity change $\Delta R/R$, and $(-\Delta\alpha)l$ at $\lambda_{\text{probe}}=406$ nm as a function of time delay ($\lambda_p=320$ nm, $F_p=2.54$ mJ/cm²). Supplementary Fig. 5(b) plots the relative absorption coefficient change $\Delta\alpha/\alpha$ at $\lambda_{\text{probe}}=406$ nm under pump fluence of 2.54 mJ/cm² and 1.91 mJ/cm². As seen, the maximum reduction in α at this wavelength is ~55%, i.e., $|\Delta\alpha/\alpha| \sim 55\%$, implying that the complete bleaching of exciton absorption occurs as the (photogenerated) carrier density reaches $\sim 2 \times 10^{20}$ cm⁻³. Supplementary Fig. 6(a) shows

the relative contribution to the absorption change, i.e., $(\frac{R}{1-R} \frac{\Delta R}{R}) / (\frac{\Delta T}{T})$, as a function of

time delay (<15 ps) at three typical wavelengths (i.e., 395 nm, 406 nm, 414 nm). As seen, the consequence of neglecting the change in reflection (ΔR) depends on the wavelength, e.g., i) at the wavelength of 406 nm, $\left| \left(\frac{R}{1-R} \frac{\Delta R}{R} \right) / \left(\frac{\Delta T}{T} \right) \right|$ is less than 0.05, indicating that ΔR of CuI thin film has negligible influence in the evaluation of the transient absorption ΔA at this probe wavelength, ii) at the longer wavelength of 414 nm, $\left(\frac{R}{1-R} \frac{\Delta R}{R} \right) / \left(\frac{\Delta T}{T} \right)$ presents relatively high positive values, up to ~ 1.25 at a time delay ~ 3.3 ps, indicating the under estimation of ΔA , iii) at the shorter wavelength of 395 nm, the $\left(\frac{R}{1-R} \frac{\Delta R}{R} \right) / \left(\frac{\Delta T}{T} \right)$ is negative, with its maximum absolute value less than ~ 0.3 , implying a little over estimation of ΔA . Supplementary Fig. 6(b) further plots the α of CuI thin film at different time delays, with the α_0 representing the absorption coefficient without pump excitation. As seen, the α is reduced at wavelength of 406 nm, while it is increased at wavelength of 395 nm or 414 nm, which is consistent with the results from Fig. 7(a), i.e., PB at ~ 406 nm and PIA at 395 or 414 nm. Therefore, we can conclude that neglecting the reflection change ΔR in our TA measurement would not qualitatively change the results. Thus, in the following sections the TA spectra are derived only from their transient transmission spectra.”

9. The authors discuss that the optically generated charge carrier density is one to two orders of magnitude larger than the calculated charge carrier density at the Mott transition, so that excitonic effects can be neglected. Recently, however, it has been shown that in ZnO, for example, the excitonic features are still present in the dielectric function even around the Mott transition, indicating that the occupation of the exciton ground state does not exceed the Mott density even at high total carrier densities (Richter et al., New Journal of Physics 22, 083066 (2020)). Moreover, it has been observed that electron-hole coupling is maintained even above the Mott transition, leading to so-called Mahan excitons (Mahan, Physical Review 153, 882 (1967) & Zimmermann, physica status solidi (b) 146, 371 (1988)). Therefore, I consider the exclusion of excitonic effects based on the calculated charge carrier density alone to be premature. In the grown samples, for example, the charge carrier density already exceeds the calculated Mott charge carrier density by an order of magnitude. Nevertheless, a clear excitonic response can be seen in the absorption spectra, indicating that excitonic effects must be expected even at high charge carrier densities. Is there any evidence of complete bleaching of the excitonic resonance in the transient transmission spectra? A selection of time-delayed transient transmission spectra seems essential and should be added, also to understand the BGR discussed later as well as the photo-induced absorption below the band gap.

Response:

Indeed, excitons exist in CuI thin film with carrier density up to $\sim 5 \times 10^{19} \text{ cm}^{-3}$, which is one order of magnitude higher than the calculated Mott density ($\sim 4.8 \times 10^{19} \text{ cm}^{-3}$), as evidenced by the exciton induced absorption peak shown in Fig. 3(a). The persistence of such excitonic feature as carrier density above the Mott density is most likely due to the formation of Mahan exciton, which was also reported in other materials with large exciton binding energies. As the hole density is over than $\sim 7 \times 10^{19} \text{ cm}^{-3}$, the exciton absorption is negligible (see Supplementary Fig. 2). From the transient transmission and the transient reflectance measurements, the transient $\Delta\alpha/\alpha$ was found to be around -55% at $\sim 406 \text{ nm}$, indicating a complete bleaching of excitonic transition as the photogenerated carrier density is $\sim 10^{20} \text{ cm}^{-3}$ (see Supplementary Fig. 6 (b)). In order to better understand the BGR effect, transient absorption spectra ($\lambda_p=320 \text{ nm}$, $F_p=0.64 \text{ mJ/cm}^2$) at different time delays (from 0.2 ps to 2 ps) are illustrated (see Supplementary Fig. 3 (a)). It is observed that the change in band gap narrowing ΔE_{BGR} induced by the BGR effect reaches its maximum value $\sim 10 \text{ meV}$ at a time delay of $\sim 1 \text{ ps}$. Meanwhile, the major PB peak exhibits a blue shift (8 meV) with increasing $|\Delta A|$ under a certain pump fluence (e.g., 0.64 mJ/cm^2). In case high carrier density filled in the band edges, long-range Coulomb screening may result in blue shift of the exciton peak (if excitons are present) due to the reduced exciton binding energy, while phase-space filling may also contribute to the blue shift of exciton peak or the absorption edge [see e.g., Baldini et al., Phys. Rev. Lett. 125, 116403 (2020)]. In addition, the maximum relative absorption coefficient $\Delta\alpha/\alpha$ was found to be ~ 0.20 at 414 nm by using the transient transmission and transient reflection measurements, further confirming an enhanced optical absorption below the bandgap in the case when photogenerated carriers are present in the CuI thin film. Accordingly, we changed the corresponding description on page 9-10 in the revised manuscript.

“However, the exciton absorption is distinct in CuI thin film with $N \sim 5.0 \times 10^{19} \text{ cm}^{-3}$ (around one order of magnitude higher than the calculated Mott density), as indicated in Fig. 3(c) and Supplementary Fig. 2. The persistence of such excitonic feature at carrier density above the Mott density is most likely due to the formation of Mahan excitons, which were also reported in other materials with large exciton binding energies[51]. As shown in Supplementary Fig. 2, as the hole density is $> \sim 7 \times 10^{19} \text{ cm}^{-3}$, the exciton absorption becomes negligible. Be noted that a fraction of EHP may be present in the photogenerated carriers. By using eq. (1), the fraction x is found to be $\sim 88\%$ and 57% for photogenerated carrier density $\sim 1 \times 10^{17} \text{ cm}^{-3}$ and $1 \times 10^{18} \text{ cm}^{-3}$, respectively. Therefore, the PB at 406 nm (or 336 nm) (see Fig. 5a, with excitation density $\sim 5.0 \times 10^{19} \text{ cm}^{-3}$) for the CuI thin film can be attributed to the formation of EHP, intra-band carrier relaxation, as well as the filling of excitonic states. In order to better understand the kinetics of PB and BGR, Supplementary Fig. 3(a, b) illustrate the TA spectra at several selected time delays ranged from 0.2 ps to 2 ps with $F_p = 0.64 \text{ mJ/cm}^2$ and $F_p = 2.54 \text{ mJ/cm}^2$, respectively. As seen, ... Under pump fluence of 0.64 (2.54) mJ/cm^2 , the change in ΔE_{BGR} due to the transient BGR effect reaches its maximum value of ~ 10 (18) meV at a time delay of ~ 1 (1.2) ps , which was estimated from the energy shift at the transition point (i.e., PIA-PB transition point, see

Supplementary Fig.3). In addition, the major PB peak exhibits a blue shift of ~ 8 meV with increasing $|\Delta A|$ under a certain pump fluence (e.g., 0.64 mJ/cm²). In case high carrier density filled in the band edges, long-range Coulomb screening may result in a blue shift of the exciton peak (if excitons are present) due to the reduced exciton binding energy, while phase-space filling may also contribute to the blue shift of exciton peak or the absorption edge[51], which is actually revealed in Supplementary Fig. 3.”

10. The increase in ΔA at a time delay of about 25 ps is attributed to the re-excitation of charge carriers by stimulated emission. However, the detection of photoluminescence would also lead to an effective increase (decrease) in transmission density (absorption) at the emission wavelength. Moreover, the discussion is based on the statement that stimulated emission was observed in CuI microwires 30 ps after optical excitation. However, Wille et al. (Applied Physics Letters 111, 031105 (2017)) show that PL emission starts almost immediately after the excitation pulse hits the sample.

Response:

This increase in ΔA at ~ 25 ps not only present at PB, but also emerges at the PIA. Actually, we noted that the so-called pump-push-probe transient spectrum (see Marcello Righetto, et al. Nature Communication, 11 (1), 2712 (2020); Hopper, T. R.; et al. ACS Energy Lett 2018, 3 (9), 2199-2205) can also present a similar feature (i.e., abrupt increase in transient transmission spectrum upon the push radiation). However, the possible spontaneous emission from the CuI is very unlikely related with such abrupt increase in ΔA at ~ 25 ps due to following reasons. Firstly, the intensity of spontaneous emission may not be high enough to cause such significant excitation. Secondly, the maximum intensity for the spontaneous emission from the CuI typically appears at a time delay less than 4 ps (see Phys. Rev. B 72, 045210, 2005), while the maximum intensity for the lasing emission from the CuI microwire was found at a time delay ~ 30 ps after laser excitation (see Applied Physics Letters 111, 031105 (2017)), which coincides with the time delay of ~ 25 ps for the abrupt increase in ΔA as observed in our CuI thin films. In addition, such increase in ΔA at a time delay of about 25 ps is only notable in case of relatively high pump fluence (e.g., $> \sim 0.1$ mJ/cm²), which is close to the lasing threshold density of ~ 0.3 mJ/cm² as found by Wille et al. (Applied Physics Letters 111, 031105 (2017)). Therefore, we believe that the increase in ΔA at a time delay of ~ 25 ps in CuI can be attributed to the re-excitation of charge carriers by EHP recombination induced stimulated emission.

11. Due to high laser fluencies, the material can be locally strongly heated, as also mentioned by the authors. It is conceivable that such heating of the material can lead to damage and even oxidation of the CuI surface in air. Are the initial absorption spectra ever fully recovered or are the changes in optical properties irreversible after pump?

Response: All the TA data are averaged from two successive measurements (one measurement includes 200 pumping excitations). Fig. R3 plots the two successive measured TA data. As seen, the data are almost identical with negligible discrepancy. This suggests that possible heating of CuI during the TA measurements would not result in noticeable damage or oxidation.

Fig. R3. TA kinetics at 406 nm from two successive measurements.

12. At a pump fluence of $> 1.5 \text{ mJ/cm}^2$, the maximum signal ΔA saturates to a constant value, possibly indicating that the excitonic absorption is indeed fully bleached at the corresponding carrier densities and that the Mott transition is thus reached here. Could the authors comment on this?

Response:

As mentioned in the manuscript, the bleaching maxima of 406 nm saturates as pump fluence $> \sim 1.5 \text{ mJ/cm}^2$, which was attributed to the EHP induced band filling and even the exhaustion of the number of available states under high excitation fluence. In order to further verify if the complete excitonic bleaching is reached at relatively high excitation density, the $\Delta\alpha/\alpha$ at 406 nm was extracted from the combination of transient reflection spectrum (i.e., $\Delta R/R$) and the transient transmission spectrum (i.e., $\Delta T/T$). Fig.R4 plots the $\Delta\alpha/\alpha$ at $\lambda_{\text{probe}}=406 \text{ nm}$ as a function of time delay ($\lambda_p=320 \text{ nm}$, $F_p=1.91$ or 2.54 mJ/cm^2). As seen, the kinetics of $\Delta\alpha/\alpha$ under these pump fluences (i.e., $F_p=1.91 \text{ mJ/cm}^2$, 2.54 mJ/cm^2) are rather close, both with a maximum absolute value of $\sim 55\%$. This clearly indicates the complete exciton bleaching under pump fluence $\geq 1.91 \text{ mJ/cm}^2$ (with excitation density $\sim 1.5 \times 10^{20} \text{ cm}^{-3}$). Taking into account the free holes (with density $\sim 5 \times 10^{19} \text{ cm}^{-3}$) in the as-grown CuI thin film, the Mott transition is actually reached as a carrier density of $\sim 2 \times 10^{20} \text{ cm}^{-3}$. Therefore, e-h plasma and (Mahan) excitons most likely coexist in the CuI thin film (with excitation density $< 1.5 \times 10^{20} \text{ cm}^{-3}$) before its recovery.

Fig. R4. $\Delta\alpha/\alpha$ at $\lambda_{\text{probe}}=406$ nm as function of time delay under pump fluences of 1.91 mJ/cm^2 and 2.54 mJ/cm^2 , respectively ($\lambda_p=320$ nm).

13. The increased absorption above the band gap is attributed to the widening of the CuI band gap with increasing temperature predicted by Xu et al. [1]. However, an increase in the band gap in CuI with increasing temperature has not been experimentally demonstrated, at least up to 300 K [2-4]. In addition, it should be noted that similar behavior above the band gap has been observed in lead halide perovskites [5], but where it was attributed only to changes in reflectivity rather than a change in absorption.

- [1] Xu et al., Journal of Physics: Condensed Matter 34, 134002 (2022)
- [2] Serrano et al., Physical Review B 65, 125110 (2002)
- [3] Krüger et al., Appl. Phys. Lett. 113, 172102 (2018)
- [4] Krüger et al., APL Materials 9, 121102 (2021)
- [5] Price et al., Nature Communications 6, 1 (2015)

Response:

As pointed out by the reviewer, there is no experimental evidence of band gap widening with increasing temperature for CuI. In fact, the bandgap of CuI was found to decrease with temperature in the temperature range of 10~300 K from existing experimental results (e.g., Krüger et al., Appl. Phys. Lett. 113, 172102 (2018)). It's very likely that the bandgap of CuI would also decrease for temperatures higher than room temperature. In order to further verify the PIA at energies larger than the bandgap, the transient $\Delta\alpha/\alpha$ at higher energies (e.g., ~395 nm) was derived from the transient transmission and the transient reflection measurements (see Fig. R5(a)). Also shown in Supplementary Fig.

6(a), the $(\frac{R}{1-R} \frac{\Delta R}{R}) / (\frac{\Delta T}{T})$ is negative with its absolute value < 0.33 , which would not

qualitatively change the result. Fig. R5(b) shows the $\Delta\alpha/\alpha$ at $\lambda_{\text{probe}}=395$ nm. It is found that the $\Delta\alpha/\alpha$ is indeed positive, indicating its absorption coefficient was enhanced after photo excitation. Such photo-induced nonresonance absorption at energies above or below the bandgap was also observed in monolayer WS_2 when excited by an intense

laser pulse [T. Jiang et al., Optics Express 26, 859 (2018)], which was ascribed to the strong photo-induced restructuring of energy bands. Accordingly, we changed the corresponding description on page 13 in the revised manuscript.

“Such photo-induced nonresonance absorption at energies above or below the bandgap was also observed in monolayer WS₂ when excited by an intense laser pulse [67], which was ascribed to the strong photo-induced restructuring of energy bands.”

Fig. R5. (a) Transmission change $\Delta T/T$, reflection change $\Delta R/R$, and $(-\Delta\alpha)l$ of CuI thin film ($N \sim 1.0 \times 10^{19} \text{ cm}^{-3}$) as function of time delay with $\lambda_{\text{probe}}=395 \text{ nm}$, $\lambda_{\text{pump}}=320 \text{ nm}$, $F_p=2.54 \text{ mJ/cm}^2$; (b) the corresponding absorption coefficient change $\Delta\alpha/\alpha$ at $\lambda_{\text{probe}}=395 \text{ nm}$.

14. For the case of two-photon induced charge carrier dynamics, the charge carrier densities were estimated to be between 4×10^{17} and $8 \times 10^{18} \text{ cm}^{-3}$. Could the author clarify how exactly the charge carrier density was calculated for two-photon excitation?

Response:

Similar to the above bandgap excitation (excitation energy $> E_g$), the photogenerated carrier density from the two-photon excitation (excitation energy $< E_g$) was also estimated with the assumption that $N_0 \propto |\Delta A|_{\text{max}}$, and using a reference value of $|\Delta A|_{\text{max}}$ from above bandgap excitation (e.g., pump wavelength of 320 nm and pump fluence of 0.64 mJ/cm^2).

Reviewer #2 (Remarks to the Author):

Optoelectronic properties and ultrafast carrier dynamics of copper iodide thin films

The proposed work puts forward a study of copper iodide (CuI) as an optoelectronic material. The study is motivated by growing interest on transparent conductors, and the

potential of CuI as a p-type transparent conductor. CuI benefits from a mobility that is higher than comparable p-type transparent conductors, although its mobility is still relatively low for envisioned/future applications. The mobility is known to be affected by defect states, amongst other mechanisms, and the proposed work studies the underlying mechanisms.

The proposed work puts forward rigorous analyses of CuI. The analyses include x-ray diffraction and Raman spectroscopy, to characterize the crystallinity, electrical measurements, to characterize the effects of growth conditions and time on the mobility, and pump-probe/transient-absorption analyses, to characterize the ultrafast relaxation in the material. The authors are to be commended for this breath of analysis. It can help resolve ongoing debates in literature on the relaxation pathways for charge carriers and excitons in CuI. The authors' characterizations of trends in photoluminescence intensity and recombination pathways versus charge carrier density were especially insightful for this. They were useful in distinguishing the combined effects of trap-assisted, bimolecular, and Auger recombination. However, a few more details should be explained or addressed in the use of equation (3). This equation plays a central role in the analysis, as it was used to estimate the excited

charge carrier density, but its validity depends upon the penetration depth of the pump beam, i.e., the reciprocal of the absorption coefficient. A large penetration depth, in comparison to the thickness of the film, could yield a substantial reflection from the back surface of the film with a corresponding enhancement to the charge carrier density. Further information is needed here for the pump wavelength, its penetration depth, and the validity of equation (3). Other than this, the authors put forward a rigorous characterization of CuI and make convincing arguments for the charge carrier dynamics within this material.

Response:

We thank the reviewer for his/her positive evaluation of our manuscript.

The pump penetration length $\delta = 1/\alpha$, which is found to be 50~60 nm at 320 nm. Therefore, our film thickness $d \sim 96$ nm ensures negligible reflection from the back surface of thin film. The equation (3), i.e., $N_0 = \frac{F_p \cdot \alpha_z}{E_p} \left(\frac{2}{1+n_z} \right)^2$, used in this paper was adopted from other published works (Hendry et al., Phys. Rev. B 76, 045214, 2007; S. Acharya, PhD thesis, Martin Luther University, 2013). This expression assumes that the optical excitation occurs in the material with a thickness $\sim \delta$ (i.e., the pump penetration length). While in the case of film thickness $d > \delta$, the excitation density may also be estimated by using $N_0 = \frac{F_p(1-R)(1-e^{-\alpha d})}{E_p d}$, where R is the reflectance at the air/CuI interface, α is the absorption coefficient at the pump wavelength. We noted that the excitation density derived from these two equations are quite close for the CuI thin films studied in this work. For instance, the excitation density N_0 ($\lambda_p = 320$ nm, $F_p =$

0.64 mJ/cm²) calculated using the former and the latter equation are 5.9×10^{19} cm⁻³ and 6.9×10^{19} cm⁻³, respectively. In addition, we also performed further measurements (e.g., the transient reflection) and analysis to consolidate our arguments in the revised manuscript.

All the changes are highlighted in red in the revised manuscript. The supplementary information is also added in the revised manuscript.

REVIEWER COMMENTS

Reviewer #1 (Remarks to the Author):

Although the authors have answered many questions satisfactorily and also shed more light on some unresolved aspects, some questions still remain. Also, at the current state of the manuscript, it is difficult to reproduce the results completely. The explicit comments can be found below.

The authors have described now in detail the model for the evaluation of the ellipsometry data. However, there is a question about the physical motivation of the chosen model functions such as Psemi-0 and Tauc-Lorenz function. In addition, the parameters obtained from the fit are not discussed. I assume that the model dielectric function (MDF) was used only to describe the line shape of the DF without further analysis. If this is the case, I think it should be briefly mentioned again in the manuscript. Also, it would be interesting to see if a numerical model (e.g. a point by point fit) would give a similar line shape, since otherwise the choice of oscillators already predetermines a particular line shape.

In addition, in the updated version of the manuscript, the authors state that an index gradient was used for the CuI layer. However, this is not explained in any detail, so neither the physical motivation for such a gradient nor its explicit modeling is discussed. Furthermore, it is unclear to which region of the gradient the absorption coefficient shown can be assigned, which greatly complicates the interpretation and understanding of the results. It also remains open whether such a gradient is similarly pronounced in all samples and possibly even changes with time, since, for example, possible adsorption of iodine on the surface was also addressed by the authors. Since evidence of such a gradient could also affect the interpretation of the transient absorption data, which is the focus of the paper, the authors should discuss it in more detail.

Furthermore, for better reproducibility, the authors should provide some information about the ellipsometer used, such as the configuration of the polarization optics. In addition, it would be useful to provide, at least in the appendix, the determined dielectric function that was used to calculate the absorption coefficient. This would also allow a better comparison with other literature results.

Finally, an explicit indication of the MSE value is not necessary, since it depends strongly on the choice of the ellipsometric parameters as well as the fit weighting.

In the updated manuscript version, the authors shed more light on the effects of the Burstein-Moss shift and the renormalization of the band gap. Although the observed shift in the PL peak associated with transitions between electrons in CB and defect states may be indicative of the effect of BGR, the quantitative analysis is not yet entirely convincing. In this regard, I have the following comments/questions:

- The bandgap is estimated by extrapolating α^2 to zero photon energy. However, due to the presence of the excitonic resonance, which is still present at carrier density of $4.16 \times 10^{19} \text{ cm}^{-3}$ (see Supplementary Fig. 2(a)), such an approach is not applicable. Although no clear excitonic peak is observed at higher carrier densities, excitonic contributions could still influence the line shape at the absorption edge. Apart from the band gap values of about 2.97 eV, which are much smaller than reported in the literature, the discussed band gap difference of 2 meV between both samples is questionable. Therefore, a more detailed analysis of the line shape of the dielectric function would be

required to analyze both the excitonic transition energy and the band gap energy in a quantitative way.

- The Burstein-Moss shift is approximated as the difference between the valence band maximum and the Fermi level ($E_v - E_F$). Could the authors explain why they do not also consider the dispersion of the conduction band, since the optical transition should satisfy $\Delta k = 0$?

- Why is the sample with $4.16 \times 10^{19} \text{ cm}^{-3}$ used as a reference for $\Delta E_{\text{BGR}} = 0$ and not another one with even lower carrier density, since the BGR might already play a role at such high carrier densities?

- The authors mention that although the absorption edge is related to the effects of BGR and BM, the PL peak is not affected by the BM shift. This statement is probably related to the fast scattering of free carriers to the CB minimum together with the assignment of the PL peak due to the transition from the CB to the Cu defect state. Could the authors elaborate on this in the manuscript?

The authors present new measurements of the transient change in reflectivity to more accurately determine the transient absorption, including the effects of the change in absorption and refractive index. Could the authors please provide either a derivation or reference to the formula presented for calculating ΔA ?

Also, the effect of refractive index change for three different wavelengths (395 nm, 406 nm, 414 nm) is presented as a function of retardation time. From this it can be seen that the effect of refractive index change is strongly dependent on wavelength and would therefore result in a different line shape of the calculated transient absorbance than if only transmission data were considered, although the calculated TA is not affected at the excitonic resonance at 406 nm. A direct comparison of the transient absorption for different cases (transmission data only, transmission and reflection data) for a given delay time would help to better understand the influence of the refractive index change.

The authors highlight that a narrowing of the band gap due to the BGR effect is observed for time delays $< 1 \text{ ps}$ (Fig. 3a), while the main PB peak associated with excitonic transitions exhibits a blue shift due to changes in the exciton binding energy as well as phase space filling effects. However, it is not clear why the shift between the PIA-PB transition point is directly related to the band gap renormalization effect. Could the authors please explain this more in detail?

The authors argue that the observed increase in ΔA at $\sim 25 \text{ ps}$ is due to the re-excitation of charge carriers by stimulated emission. One of the main arguments is that the observation of laser emission occurs with a time delay of 30 ps after laser excitation (see Applied Physics Letters 111, 031105 (2017)). Unfortunately, this statement is not correct. Fig. 3a of the above reference shows the time-resolved PL spectra of a CuI above the laser threshold. Although the emission starts at about 30 ps according to the vertical scale, the horizontal dashed white line indicates the impingement of the excitation pulse, which is also at $\sim 30 \text{ ps}$. Thus, the time delay between the onset of laser emission and laser excitation is nearly zero, and the 30 ps is more like an overall system delay. Thus, based on the presented data, the statement about the stimulated emission is quite moderate.

In general, the question arises whether possible effects due to emission must also be considered when discussing the transient absorption spectra (even a few ps after excitation). It would therefore be important to clarify whether the transient absorption spectra presented are corrected for the PL signal, which may be detected simultaneously with the transmitted light and thus mimic absorption bleaching

in a particular spectral region.

Some minor points:

At some points, there are currently still missing values for the axes labels, e.g.: Fig.4 (a-c), Fig. 8 (a), Fig. 11 (b)

The authors might consider using "arb.unit" instead of "a.u." to avoid confusion with, for example, atomic units.

Reviewer #2 (Remarks to the Author):

The authors' replies and revisions to the manuscript are satisfactory. In this reviewer's opinion, the revised manuscript is suitable for publication.

REVIEWER COMMENTS

Reviewer #1 (Remarks to the Author):

Although the authors have answered many questions satisfactorily and also shed more light on some unresolved aspects, some questions still remain. Also, at the current state of the manuscript, it is difficult to reproduce the results completely. The explicit comments can be found below.

Response:

We thank the reviewer for his/her careful review of our revised manuscript. In the following, we will answer these questions raised.

1. The authors have described now in detail the model for the evaluation of the ellipsometry data. However, there is a question about the physical motivation of the chosen model functions such as Psemi-0 and Tauc-Lorenz function. In addition, the parameters obtained from the fit are not discussed. I assume that the model dielectric function (MDF) was used only to describe the line shape of the DF without further analysis. If this is the case, I think it should be briefly mentioned again in the manuscript. Also, it would be interesting to see if a numerical model (e.g. a point by point fit) would give a similar line shape, since otherwise the choice of oscillators already predetermines a particular line shape.

Response:

In the revised manuscript, we used the first derivative of $d\varepsilon_2/dE$ to extract the optical band gaps for the CuI thin films with different hole densities. For the SE data fitting, we also tried the point-by-point fit. Although similar line shape of dielectric function was obtained by using the point-by-point fit, there is notable difference in the relatively low energy range (1~3 eV), which might be due to the fact that the point-by-point fit used in our modeling does not satisfy the Kramers-Kronig consistency. Nevertheless, we noted that the obtained dielectric function (see the Supplementary Fig. 2(a)) for the as-grown CuI thin film is quite close to the that of CuI thin film as determined numerically by a Kramers-Kronig consistent point-by-point regression analysis (see Kruger et al, Appl. Phys. Lett. 113, 172102, 2018). Accordingly, we revised the corresponding description on page 6 in the revised manuscript.

“We find that the derived dielectric function of as-grown CuI thin film (see Supplementary Fig. 2(a)) is quite close to that of CuI film with similar hole density ($\sim 1.6 \times 10^{19} \text{ cm}^{-3}$) as determined numerically by a Kramers-Kronig consistent point-by-point regression analysis [28]”

2. In addition, in the updated version of the manuscript, the authors state that an index gradient was used for the CuI layer. However, this is not explained in any detail, so neither the physical motivation for such a gradient nor its explicit modeling is discussed. Furthermore, it is unclear to which region of the gradient the absorption coefficient

shown can be assigned, which greatly complicates the interpretation and understanding of the results. It also remains open whether such a gradient is similarly pronounced in all samples and possibly even changes with time, since, for example, possible adsorption of iodine on the surface was also addressed by the authors. Since evidence of such a gradient could also affect the interpretation of the transient absorption data, which is the focus of the paper, the authors should discuss it in more detail.

Response:

Since the CuI thin films exhibit distinct graded microstructures, it is necessary to use a proper index grading in the optical model to describe the variation of refractive index throughout the film thickness induced by its graded microstructures. Actually, we found that the SE data can hardly be fitted well without using the index grading. For simplicity, the linear variation of refractive index throughout the thickness of CuI thin film was assumed. It is found that the variation of optical properties from the bottom of the film to the film surface is quite low (e.g., the change in the refractive index $\Delta n/n < 5\%$, while the change in the absorption coefficient $\Delta\alpha/\alpha < 6.4\%$) in the spectral range of 3~4 eV. Such small change in the index would not affect the main conclusion. In the manuscript, the mean value of the optical constants (e.g., refractive index n , absorption coefficient α) of the thin film were used for calculations and discussions. Generally, such gradients are present in most of these samples. We also found that PTA treatments used in this work can slightly change the gradient due to the change in the microstructure of the film, while aging in the ambient air did not show noticeable effect on such gradient. We believe that such small index grading may not significantly affect the interpretation of the transient absorption data in this work. Accordingly, we revised the corresponding description on page 6 in the revised manuscript.

“Be noted that index grading was used for the CuI layer in the optical model to describe the linear variation of refractive index throughout the film thickness induced by its graded microstructure as frequently observed in transparent conducting thin films (e.g., tin doped indium oxide) [39]. It is found that the variation of optical properties from the bottom of the film to the film surface is quite low (e.g., the change in the refractive index $\Delta n/n < 5\%$, while the change in the absorption coefficient $\Delta\alpha/\alpha < 6.4\%$) in the spectral range of 3~4 eV. Such small change in the index would not affect the main conclusion. In the following context, the mean value of the optical constants (e.g., dielectric function, refractive index n , absorption coefficient α) of the thin film were used for calculations and discussions.”

3. Furthermore, for better reproducibility, the authors should provide some information about the ellipsometer used, such as the configuration of the polarization optics. In addition, it would be useful to provide, at least in the appendix, the determined dielectric function that was used to calculate the absorption coefficient. This would also allow a better comparison with other literature results.

Finally, an explicit indication of the MSE value is not necessary, since it depends strongly on the choice of the ellipsometric parameters as well as the fit weighting.

Response:

The ellipsometer used for the SE measurements is a commercial rotating-compensator ellipsometer (Eoptics, SE-VM-L) in the polarizer-compensator-sample-compensator-analyzer (PCSCA) configuration. The derived dielectric function of as-grown CuI thin film was added in the Supplementary Fig. 2(a). Also, the indication of MSE value was deleted in the revised manuscript. Accordingly, we added the corresponding description on page 23 and page 6 in the revised manuscript.

“using a commercial rotating-compensator ellipsometer (Eoptics, SE-VM-L) in the polarizer-compensator-sample-compensator-analyzer (PCSCA) configuration.

“We find that the derived dielectric function of as-grown CuI thin film (see Supplementary Fig. 2(a)) is quite close to that of CuI film with similar hole density ($\sim 1.6 \times 10^{19} \text{ cm}^{-3}$) as determined numerically by a Kramers-Kronig consistent point-by-point regression analysis [28].”

4. In the updated manuscript version, the authors shed more light on the effects of the Burstein-Moss shift and the renormalization of the band gap. Although the observed shift in the PL peak associated with transitions between electrons in CB and defect states may be indicative of the effect of BGR, the quantitative analysis is not yet entirely convincing. In this regard, I have the following comments/questions:

- The bandgap is estimated by extrapolating α^2 to zero photon energy. However, due to the presence of the excitonic resonance, which is still present at carrier density of $4.16 \times 10^{19} \text{ cm}^{-3}$ (see Supplementary Fig. 2(a)), such an approach is not applicable. Although no clear excitonic peak is observed at higher carrier densities, excitonic contributions could still influence the line shape at the absorption edge. Apart from the band gap values of about 2.97 eV, which are much smaller than reported in the literature, the discussed band gap difference of 2 meV between both samples is questionable. Therefore, a more detailed analysis of the line shape of the dielectric function would be required to analyze both the excitonic transition energy and the band gap energy in a quantitative way.

Response:

Indeed, the optical band gap E_G^{opt} derived by extrapolating α^2 to zero photo energy is underestimated mainly due to the excitonic resonance. In order to get more reasonable values for the E_G^{opt} of CuI thin films with different hole densities, the E_G^{opt} was estimated from the zero crossing of first-derivation of ϵ_2 , $d\epsilon_2/dE$ (Ref. M. Grundmann et al., Appl. Phys. Lett. 113, 172102, 2018). It is found that the obtained E_G^{opt} gradually increases from ~ 3.057 eV to ~ 3.11 eV as the hole density N increases from $2.79 \times 10^{17} \text{ cm}^{-3}$ to $1.68 \times 10^{20} \text{ cm}^{-3}$ (see Supplementary Fig. 2(d)). As N is less than $5 \times 10^{19} \text{ cm}^{-3}$, the obtained E_G^{opt} is approximately equal to the corresponding free exciton peak

energy (see the green region of Supplementary Fig. 2(d)), while as N is larger than $\sim 7 \times 10^{19} \text{ cm}^{-3}$, the E_G^{opt} is close to its band edge absorption energy (see the red region of Supplementary Fig. 2(d)). This implies that exciton peak energy blue shifts as increasing hole density in CuI thin film. However, the E_G^{opt} cannot be expressed by $E_G^{\text{opt}} = E_G + \Delta E_{\text{BM}} - \Delta E_{\text{BGR}}$ in case of presence of distinct excitonic absorption, since the obtained E_G^{opt} is even smaller than the bandgap E_G of 3.1 eV. In our calculation, we found that $\Delta E_{\text{BM}}=0$ as $N < \sim 7.6 \times 10^{19} \text{ cm}^{-3}$. The ΔE_{BGR} for CuI thin film with $N \sim 1.68 \times 10^{20} \text{ cm}^{-3}$ is $\sim 48 \text{ meV}$ (with consideration of the dispersion of conduction band as well), which is larger than the energy shift (i.e., 21 meV) as observed in the PL spectra (Fig. 4b). This is understandable, since the 48 meV might be the upper limit of ΔE_{BGR} for our CuI thin films, while the energy shift of 21 meV is more related to the change in the ΔE_{BGR} for the sample with increasing aging time. Since the ΔE_{BGR} is determined indirectly (i.e., by using $\Delta E_{\text{BGR}} = E_G + \Delta E_{\text{BM}} - E_G^{\text{opt}}$), quantification of the BGR effect in CuI thin films with distinct excitonic resonance (i.e., samples with $N < \sim 7.6 \times 10^{19} \text{ cm}^{-3}$) is not achieved in the present work. Accordingly, we revised the corresponding description on page 7-8 in the revised manuscript.

“The optical band gap E_G^{opt} was estimated from the zero crossing of the first-derivative of the ε_2 spectra, $d\varepsilon_2/dE$ (see the inset of Supplementary Fig. 2(d))[28]. As the hole density increases from $2.79 \times 10^{17} \text{ cm}^{-3}$ to $1.68 \times 10^{20} \text{ cm}^{-3}$, the E_G^{opt} of CuI thin film increases from $\sim 3.057 \text{ eV}$ to $\sim 3.11 \text{ eV}$, as shown in the Supplementary Fig. 2(d). As $N < \sim 5 \times 10^{19} \text{ cm}^{-3}$, the obtained E_G^{opt} approximately equal to the corresponding free exciton peak energy (see the green region of Supplementary Fig. 2(d)), while as $N > \sim 7 \times 10^{19} \text{ cm}^{-3}$, the E_G^{opt} is close to its band edge absorption energy (see the red region of Supplementary Fig. 2(d)). This implies that exciton peak energy blue shifts as the hole density in CuI thin film increases, mainly due to the enhanced Coulomb screening induced reduction in exciton binding energy (E_b), similar to that observed in anatase TiO_2 [52]. The energy difference between the valence band edge E_V and the Fermi level E_F (i.e., $E_V - E_F$) was calculated by using $N = 2 \frac{m_h k_B T^{3/2}}{2\pi \hbar^2} F_{1/2}\left(-\frac{E_F - E_V}{k_B T}\right)$, where N is the hole density in the valence band, m_h is the density of state hole effective mass ($m_h = (m_{hh}^{3/2} + m_{lh}^{3/2})^{2/3}$) which is found to be around $2.47 m_e$ for CuI [16], k_B is the Boltzmann constant, T is the temperature, \hbar is the Plank constant, $F_{1/2}$ is the

Fermi-Dirac integral which was calculated by method proposed by Bednarczyk et al.[46]. It is found that E_V-E_F is around 0, and 29 meV for samples with hole density $N\sim 7.6\times 10^{19}$ and $1.68\times 10^{20}\text{cm}^{-3}$, respectively. Given the similar band dispersions (e.g., $m_e^ = m_h^* = 0.3m_0$) for the CBM and the VBM of CuI [16], the Burstein-Moss (BM) shift (ΔE_{BM}) due to the hole filling in the valence band of CuI thin film is approximately equal to the $2(E_V-E_F)$, i.e., $\Delta E_{BM} \sim 2(E_V-E_F)$, indicating that the ΔE_{BM} of CuI thin film with $N\sim 1.68\times 10^{20}\text{cm}^{-3}$ is ~ 58 meV. For CuI thin film with negligible exciton resonance, the optical band gap can be expressed as*

$$E_G^{\text{opt}} = E_G + \Delta E_{BM} - \Delta E_{BGR}, \quad (1)$$

where E_G is the intrinsic band gap of 3.1 eV, ΔE_{BGR} is the band gap narrowing due to the band gap renormalization (BGR). It is found that the ΔE_{BGR} for CuI thin film with $N\sim 1.68\times 10^{20}\text{cm}^{-3}$ is ~ 48 meV, which is larger than the energy shift (i.e., 21 meV) as observed in the PL spectra (Fig. 4b). This is understandable, since the 48 meV represents the upper limit of ΔE_{BGR} for our CuI thin films, while the energy shift of ~ 21 meV is related to the change in the ΔE_{BGR} (i.e. $\Delta(\Delta E_{BGR})$) for the sample with increasing aging time. Since the ΔE_{BGR} is determined indirectly from Eq. (1), quantification of the BGR effect in CuI thin films with distinct excitonic resonance (i.e., samples with $N\sim 7.6\times 10^{19}\text{cm}^{-3}$) is not achieved in the present work. The right Y axis of Supplementary Fig. 2(d) further illustrates the quantity of E_b' (i.e., $E_b' = 3.1 - E_G^{\text{opt}}$) for rough estimates of exciton binding energy E_b as a function of N with the assumption of $\Delta E_{BGR} = 0$. Apparently, E_b' should be slightly larger than the E_b ($E_b \approx 3.1 - \Delta E_{BGR} - E_G^{\text{opt}}$), and the negative value of E_b' indicates the negligible excitonic resonance.”

5. The Burstein-Moss shift is approximated as the difference between the valence band maximum and the Fermi level (E_v-E_F). Could the authors explain why they do not also consider the dispersion of the conduction band, since the optical transition should satisfy $\Delta k=0$?

Response:

Indeed, the optical transition should satisfy $\Delta k=0$. We are sorry for such neglect. In the revised manuscript, the dispersion of conduction band was also taken into account. For the CuI, the conduction band minimum (CBM) and the valence band maximum (VBM) exhibit similar band dispersion. The electron effective mass m_e^* and the hole effective mass m_h^* at the band edges are equal, i.e., $m_e^* = m_h^* = 0.3 m_0$ (Grundmann et al., Phys. Status Solidi A 210, 1671, 2013). Given the similar band dispersion for the CBM and the VBM of CuI, the corrected ΔE_{BM} should be close to double of (E_V-E_F) , i.e., $\Delta E_{BM} \sim 2(E_V-E_F)$. Then the ΔE_{BM} and the ΔE_{BGR} for CuI with $N\sim 1.68\times 10^{20}\text{cm}^{-3}$ is found to be ~ 58 meV and ~ 48 meV, respectively.

6. Why is the sample with $4.16 \times 10^{19} \text{ cm}^{-3}$ used as a reference for $\Delta E_{\text{BGR}}=0$ and not another one with even lower carrier density, since the BGR might already play a role at such high carrier densities?

Response:

In the revised manuscript, we calculated the ΔE_{BGR} for the CuI thin film with relatively high hole density by using $\Delta E_{\text{BGR}} = E_{\text{G}} + \Delta E_{\text{BM}} - E_{\text{G}}^{\text{opt}}$, without resorting to any reference sample. However, as mentioned earlier, the estimation of ΔE_{BGR} is not achieved for CuI thin films with notable excitonic resonance in the present work.

7. The authors mention that although the absorption edge is related to the effects of BGR and BM, the PL peak is not affected by the BM shift. This statement is probably related to the fast scattering of free carriers to the CB minimum together with the assignment of the PL peak due to the transition from the CB to the Cu defect state. Could the authors elaborate on this in the manuscript?

Response:

In our steady-state PL measurements, we found that the PL spectra are dominated by the (e, V_{Cu}^0) recombination in the CuI thin films with hole densities up to $\sim 10^{20} \text{ cm}^{-3}$, without any noticeable effect from the BM shift in the degenerate samples. This can be caused by following two reasons, i) fast non-radiative recombination (e.g., Auger recombination, SRH recombination) of the majority photogenerated carriers as discussed in the later sections; ii) efficient radiative (e, V_{Cu}^0) recombination process related with the photogenerated electrons relaxed to the CBM by fast scattering and the prominence of V_{Cu}^0 defect states. While these two types of ultrafast recombination (i.e., non-radiative and radiative) processes are probably unrelated with the BM shift in the CuI thin films. Accordingly, we added the corresponding description on page 8 in the revised manuscript.

“The independence of PL peak on the BM shift of CuI thin film was revealed in our experiments. As seen in Fig. 4(b), the PL spectra are dominated by the (e, V_{Cu}^0) recombination in the CuI thin films with hole densities up to $\sim 10^{20} \text{ cm}^{-3}$, without any noticeable effect from the BM shift in the degenerate samples. This can be caused by following two reasons, i) fast non-radiative recombination (e.g., Auger recombination, SRH recombination) of the majority photogenerated carriers as discussed in the later sections; ii) efficient radiative (e, V_{Cu}^0) recombination process related to the photogenerated electrons relaxed to the CBM by fast scattering and the prominence of V_{Cu}^0 defect states. These two types of ultrafast recombination (i.e., non-radiative and radiative) processes are probably unrelated with the BM shift in the CuI thin films.”

8. The authors present new measurements of the transient change in reflectivity to more accurately determine the transient absorption, including the effects of the change in absorption and refractive index. Could the authors please provide either a derivation or reference to the formula presented for calculating ΔA ?

Response:

The derivation of the transient absorption coefficient change $\Delta\alpha$ was provided in the supplementary information (Supplementary Note 1) of the revised manuscript. Accordingly, we added the description on page 15 in the revised manuscript.

“The derivation of Eq. (11) can be found in the Supplementary Note 1.”

9. Also, the effect of refractive index change for three different wavelengths (395 nm, 406 nm, 414 nm) is presented as a function of retardation time. From this it can be seen that the effect of refractive index change is strongly dependent on wavelength and would therefore result in a different line shape of the calculated transient absorbance than if only transmission data were considered, although the calculated TA is not affected at the excitonic resonance at 406 nm. A direct comparison of the transient absorption for different cases (transmission data only, transmission and reflection data) for a given delay time would help to better understand the influence of the refractive index change.

Response:

In the revised manuscript, we provide a direct comparison of transient absorption spectra (see the Supplementary Fig. 6(c)) with and without consideration of transient reflection change. Accordingly, we added the corresponding description on page 15 in the revised manuscript.

“A direct comparison of the transient absorption spectra without ($\Delta T/(T+R)$) and with ($(\Delta R+\Delta T)/(T+R)$) consideration of transient reflection change ($\Delta R/(T+R)$) at a time delay of 2 ps are depicted in Supplementary Fig. 6(c). Again, it is found that the effect of ΔR on the transient absorption spectra is not significant.”

10. The authors highlight that a narrowing of the band gap due to the BGR effect is observed for time delays <1 ps (Fig. 3a), while the main PB peak associated with excitonic transitions exhibits a blue shift due to changes in the exciton binding energy as well as phase space filling effects. However, it is not clear why the shift between the PIA-PB transition point is directly related to the band gap renormalization effect. Could the authors please explain this more in detail?

Response:

The PIA in the spectral range of ~410 nm to ~440 nm (see Supplementary Fig. 3(a-b)) is related to the change of BGR effect in the CuI thin film, while the PIA-PB transition point denotes the photon wavelength above which the ΔA becomes a positive value. With increasing carrier density in the band edges, the band gap would decrease due to BGR effects, resulting in a redshift in the absorption edge. Such redshift in the absorption edge would cause a similar redshift of the PIA-PB transition point, as shown in the Supplementary Fig. 3(a). Therefore, the PIA-PB transition point is an indication of change in the BGR effect.

11. The authors argue that the observed increase in ΔA at ~25 ps is due to the re-excitation of charge carriers by stimulated emission. One of the main arguments is that the observation of laser emission occurs with a time delay of 30 ps after laser excitation (see Applied Physics Letters 111, 031105 (2017)). Unfortunately, this statement is not correct. Fig.3a of the above reference shows the time-resolved PL spectra of a CuI above the laser threshold. Although the emission starts at about 30 ps according to the vertical scale, the horizontal dashed white line indicates the impingement of the excitation pulse, which is also at ~30 ps. Thus, the time delay between the onset of laser emission and laser excitation is nearly zero, and the 30 ps is more like an overall system delay. Thus, based on the presented data, the statement about the stimulated emission is quite moderate. In general, the question arises whether possible effects due to emission must also be considered when discussing the transient absorption spectra (even a few ps after excitation). It would therefore be important to clarify whether the transient absorption spectra presented are corrected for the PL signal, which may be detected simultaneously with the transmitted light and thus mimic absorption bleaching in a particular spectral region.

Response:

In order to clarify the time delay as indicated in the Fig. 3(a) of this reference (Applied Physics Letters 111, 031105 (2017)), we communicated with one of the authors of this reference. The author said that the time resolution of their PL spectrometer has been stated as 20 ps and no effort was made to investigate the time delay of PL peak with regard to the excitation. In this condition, as pointed out by the reviewer, the ~30 ps is more like an overall system delay. Nevertheless, we still believe that the increase in ΔA at a time delay of ~25 ps in CuI is most likely attributed to the re-excitation of charge carriers by EHP recombination induced stimulated emission. Therefore, we revised the expression by deleting the statement “with the maximum emission intensity appeared at ~30 ps after excitation” on page 12 in the revised manuscript.

Some minor points:

At some points, there are currently still missing values for the axes labels, e.g.: Fig.4 (a-c), Fig. 8 (a), Fig. 11 (b)

The authors might consider using "arb.unit" instead of "a.u." to avoid confusion with, for example, atomic units.

Response:

We corrected these issues in the revised manuscript.

Reviewer #2 (Remarks to the Author):

The authors' replies and revisions to the manuscript are satisfactory. In this reviewer's opinion, the revised manuscript is suitable for publication.

Response:

We thank the reviewer for his/her careful review of our revised manuscript.

All the changes are highlighted in red in the revised manuscript.

REVIEWERS' COMMENTS

Reviewer #1 (Remarks to the Author):

N/A